# Analysis of the genomic landscape of yolk sac tumors reveals mechanisms of evolution and chemoresistance

Xuan Zong [1,5], Ying Zhang[1,5], Xinxin Peng[2,5], Dongyan Cao[1], Mei Yu[1], Jinhui Wang[1], Hongyue Li[2], Xuejiao Guo[2], Han Liang [3,4 ✉] & Jiaxin Yang [1 ✉]

Yolk sac tumors (YSTs) are a major histological subtype of malignant ovarian germ cell tumors with a relatively poor prognosis. The molecular basis of this disease has not been thoroughly characterized at the genomic level. Here we perform whole-exome and RNA sequencing on 41 clinical tumor samples from 30 YST patients, with distinct responses to cisplatin-based chemotherapy. We show that microsatellite instability status and mutational signatures are informative of chemoresistance. We identify somatic driver candidates, including significantly mutated genes *KRAS* and *KIT* and copy-number alteration drivers, including deleted *ARID1A* and *PARK2*, and amplified *ZNF217*, *CDKN1B*, and *KRAS*. YSTs have very infrequent *TP53* mutations, whereas the tumors from patients with abnormal gonadal development contain both *KRAS* and *TP53* mutations. We further reveal a role of *OVOL2* overexpression in YST resistance to cisplatin. This study lays a critical foundation for understanding key molecular aberrations in YSTs and developing related therapeutic strategies.

[1] Department of Obstetrics and Gynecology, Peking Union Medical College Hospital, Chinese Academy of Medical Sciences and Peking Union Medical College, Beijing, China. [2] Precision Scientific (Beijing) Co., Ltd., Beijing, China. [3] Department of Bioinformatics and Computational Biology, The University of Texas MD Anderson Cancer Center, Houston, TX, USA. [4] Department of Systems Biology, The University of Texas MD Anderson Cancer Center, Houston, TX, USA. [5]These authors contributed equally: Xuan Zong, Ying Zhang, Xinxin Peng. ✉email: hliang1@mdanderson.org; yangjiaxin@pumch.cn

Yolk sac tumors (YSTs) are the second most common histological subtype of malignant ovarian germ cell tumors (MOGCTs) that account for 3% of all ovarian tumors and usually affect adolescents and young adults[1]. YSTs are characterized by highly elevated levels of serum alpha-fetoprotein (AFP)[2]. Histologically, a vast majority of YSTs present areas with primitive extraembryonal morphology and reticular–microcystic pattern, and this is the most frequent and diagnostic feature[3]. Since the Schiller–Duval bodies are morphologically similar to endodermal sinuses in the murine placenta, YST is also known as endodermal sinus tumor[4]. Similar to other MOGCTs, the standard management for patients with YST is surgery, followed by adequate cisplatin-based chemotherapy. Although >80% of patients with MOGCT can achieve long-term survival, the prognosis of patients with YST is less favorable compared to other subtypes[5]. Treatment options for YST recurrence are very limited, and >50% of the relapsed patients die of disease even after cytoreductive surgery and salvage chemotherapy[6]. Chemoresistance is the main reason for the refractoriness. Therefore, it is clinically important to elucidate the molecular mechanisms associated with the YST resistance to chemotherapy.

The current understanding of YSTs at the molecular level is very limited, and is mostly derived from testicular germ cell tumors (TGCTs), the male counterpart of MOGCTs. Both MOGCTs and TGCTs derive from primordial germ cells and have similar onset ages, histological subtypes, and treatment responses[7]. TGCTs are characterized by low somatic mutation rates and high frequencies of somatic copy-number alterations (SCNAs)[8,9]. *KIT* and *KRAS* are the two most frequently mutated genes[10,11]. Distinct from almost all other solid tumors, *TP53* mutations are rare in TGCTs, and an intact p53 status is considered as an important mechanism of chemo-sensitivity because it induces a strong apoptotic response to DNA damage[12]. However, YSTs have some specific molecular features that are distinct from other MOGCTs and TGCTs. For example, compared to dysgerminoma and GCTs, YSTs show distinct overexpression of genes involved in WNT/β-catenin and TGF-β/BMP pathways[13,14]. Despite recent cancer genomic characterization efforts, YSTs are among the poorly characterized cancer types at the molecular level. The genomic landscape, evolutionary pattern, and chemoresistance-related mechanisms of this disease are largely unknown, resulting in a critical knowledge gap for developing diagnostic, prognostic, and therapeutic strategies. Here we perform whole-exome and RNA-sequencing on a large collection of clinical YST samples to systematically identify molecular drivers and key features related to chemoresistance in this disease.

## Results

### Somatic mutation rate and microsatellite stability status.
To characterize the genomic landscape of YSTs, we recruited 30 patients with YST and obtained tumor and blood samples. We performed whole-exome sequencing on the tumor and blood samples, which included 26 primary and 15 relapsed YST samples (four patients did not have primary samples, seven patients had both primary and relapsed samples, among which one had multiple relapsed samples, Fig. 1A, Supplementary Data 1). Most patients had received the standard treatment, i.e., surgery followed by cisplatin-based chemotherapy. We classified the 30 recruited patients into chemo-sensitive ($n = 20$) and chemo-resistant ($n = 10$) groups based on whether there was a relapse within six months after the initial treatment (Fig. 1A). AFP is a standard tumor marker for diagnosis and treatment evaluation in YSTs[15]. Indeed, we found that after six months of the initial treatment, the AFP level was significantly higher in the chemo-resistant

group than in the sensitive group (t-test, $p = 3.2 \times 10^{-4}$, Fig. 1B), further confirming our response group classification.

Based on whole-exome sequencing data (mean depth: tumor 243×, normal 139×), we employed a multiple-caller-based MC3 approach to call single nucleotide variants (SNVs) and small indels. The MC3 method is more accurate than a single caller and was recently employed to generate the high-quality mutation data of The Cancer Genome Atlas (TCGA), across 33 cancer types in a consistent way[16]. Across the 41 YST samples of the 30 patients, we identified an average of 878 SNVs (range: 244–5326) and 36 indels (range: 13–92) (Supplementary Data 2). Compared with other cancer types, YSTs exhibited a moderate level of tumor mutation burden (TMB), which was significantly lower than that of ovarian cancer (OV) (Wilcoxon rank-sum test, $p = 4.2 \times 10^{-4}$) but significantly higher than that of TGCT or its subtypes (Wilcoxon rank-sum test, TGCT vs. YST: $p < 4.2 \times 10^{-10}$, TGCT_nonseminoma vs. YST: $p = 5.5 \times 10^{-10}$; TGCT_seminoma vs. YST: $p = 3.8 \times 10^{-10}$) (Fig. 1C). We found that the TMB level showed a significant positive correlation with age at diagnosis (Spearman's rank test, $p = 10^{-3}$, Fig. 1D), presumably reflecting the mutation accumulation effect over time. We also examined the correlation of TMB with the chemotherapy response and found no significant difference between the chemo-sensitive and chemo-resistant groups (Supplementary Fig. 1). Compared with matched primary tumors, relapsed tumors had much higher TMB (paired Wilcoxon rank-sum test, $p = 5 \times 10^{-3}$, Fig. 1E), which may be partially due to the exposure of chemotherapy. Finally, we examined the distribution of microsatellite instability (MSI) scores in the 26 primary YSTs using MANTIS[17]. Interestingly, we found that the MSI score was significantly higher in the resistant group than in the sensitive group (Wilcoxon rank-sum test, $p = 0.03$, Fig. 1F). Since MSI-high has been established as a biomarker for chemoresistance in colon cancer[18], our results suggest that MSI status may also serve as a predictive marker of resistance to chemotherapy for patients with YST.

### Mutational signatures.
Mutational signatures of a tumor provide valuable information on the underlying mutational process operating on the cancer genome[19]. We, therefore, characterized the mutational spectrum of six base substitutions in all primary tumor samples (Fig. 2A and Supplementary Fig. 2A) and compared it with those of TGCT and OV, and found that YSTs exhibited fewer C > A/G > T mutations but many more T > C/A > G mutations (Supplementary Fig. 2B). To investigate the mutational signature of YSTs, we decomposed the mutational spectrum using SignatureAnalyzer[20], a widely-used Bayesian nonnegative matrix factorization algorithm-based mutational signature profiling tool and identified three signatures, W1, W2, and W3 (Fig. 2B and Supplementary Fig. 2C). Cosine similarity analysis indicated that W1 and W3 largely corresponded to the annotated COSMIC signature 23 (0.74; etiology, unknown) and signature 26 (0.77; etiology, associated with defective DNA mismatch repair); while W2 appeared to be a mixture of signature 6 (0.86, etiology, associated with defective DNA mismatch repair) and signature 1 (0.76, etiology, associated with age of cancer diagnosis), respectively (Supplementary Fig. 2D).

To examine the potential clinical utility of these de novo mutational signatures, we next examined their relative contributions among three YST sample groups: sensitive primary tumors, resistant primary tumors, and relapsed tumors (Fig. 2C). Signature W1 showed much higher contributions in relapsed tumors than primary tumors (Kruskal–Wallis test, $p = 1.4 \times 10^{-5}$, FDR = $4.2 \times 10^{-5}$), and among the primary tumors, its contribution was significantly higher in sensitive tumors than resistant

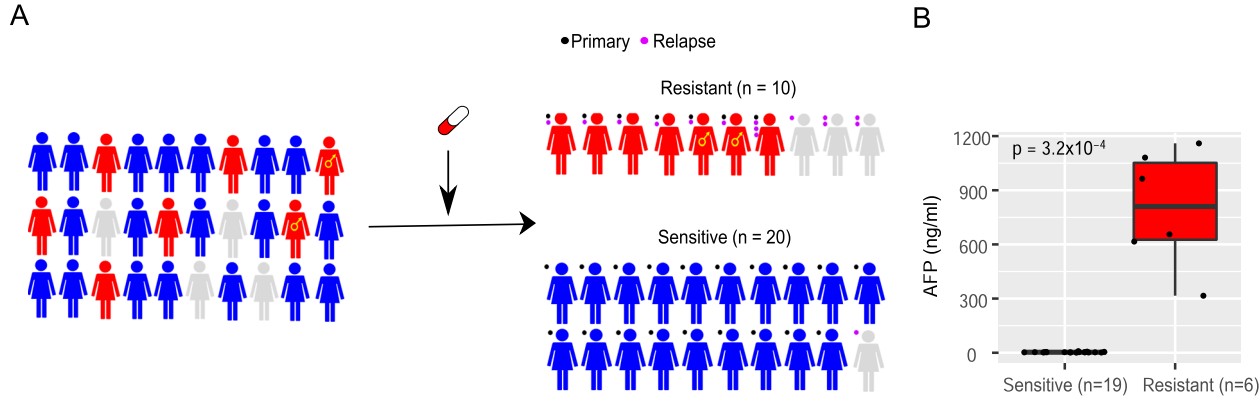

tumors (Wilcoxon rank-sum test, $p = 0.035$, FDR $= 0.1$). Signature W2 showed a clear decreasing trend from sensitive primary tumors, to resistant primary tumors, to relapsed tumors (Kruskal–Wallis test, $p = 2.1 \times 10^{-3}$; FDR $= 3.2 \times 10^{-3}$) (Fig. 2C). We also examined the differences among the three tumor groups using the established COSMIC signatures and identified five significantly differential signatures (1, 3, 4, 6, and 23, Supplementary Fig. S2E). In particular, the contribution of signature 1

was significantly higher in resistant tumors than sensitive tumors (Wilcoxon rank-sum test, $p = 0.015$, FDR $= 0.17$); the contributions of signature 4 (etiology, exposure of tobacco mutagens) and signature 23 (etiology, unknown) were much higher in relapsed tumors than primary tumors, and this pattern may reflect the effect of chemotherapy exposure. These results suggest that YST mutational signatures are also informative of tumor response prediction.

**Fig. 1 The overview study design and tumor mutation burden of YSTs. A** Study design. This study includes 41 YST samples from 30 patients. Based on the response to chemotherapy, patients were classified into chemo-resistant (n = 10) and chemo-sensitive groups (n = 20). Of note, two YST patients with 46 XY pure gonadal dysgenesis are marked with the male symbol. Primary and relapsed tumor samples are labeled as black and purple dots, respectively. **B** The AFP level comparison of chemo-sensitive and -resistant patients whose primary tumors were sequenced six months after initial treatment. Two-sided Wilcoxon rank sum test was used to compute the p value (sensitive: n = 19 biologically independent samples; resistant: n = 6). **C** The TMB distribution of YSTs in the context of 33 TCGA cancer types. YST and two related cancer types, OV and TGCT, are highlighted in red boxes. **D** The correlation between TMB and age at diagnosis in YSTs. The blue line and the shaded area represent the regression line and the related 95% confidence interval, respectively. Spearman rank correlation between age and TMB is shown. **E** The TMB difference between matched primary and relapsed tumor samples. The dotted lines indicate the matched sample pairs. For patients with multiple relapse samples, the mean TMB values were used. Two-sided t-test was used to compute the p value (n = 7 samples). **F** The microsatellite instability (MSI) scores between resistant and sensitive primary tumors. Two-sided Wilcoxon rank sum test was used to compute the p value (primary_sensitive: n = 19; primary_resistant: n = 7). **C, D, F** Only primary tumors are included. In the box plots, the middle line in the box is the median, the bottom, and top of the box are the first and third quartiles, and the whiskers extend to 1.5× interquartile range of the lower and the upper quartiles, respectively.

**Significantly mutated and somatic copy-number altered genes.** To investigate mutation driver genes in YSTs, we employed MuSiC2[21] to identify significantly mutated genes (SMGs) based on the observed mutation frequency relative to the background expectation. We included 24 primary YST samples from patients with normal gonadal development for this analysis (two primary tumor samples from patients with gonadal dysgenesis were excluded from this analysis and were analyzed separately). Among the top 16 SMGs identified (FDR < 0.2, Supplementary Table 1), KRAS and KIT are known drivers in seminomas TGCT and many cancer types[10,11,21] (Fig. 3A). Among the remaining 14 SMGs, the significance of ZNF708, FRG1, and NPAS1 was confirmed by another tool, MutSigCV[22], suggesting them to be drivers. Interestingly, the most famous cancer gene, TP53, was not mutated in any of these YST samples, in sharp contrast to its high mutation frequency in other cancer types, especially ovarian cancer, which has the second-highest frequency of TP53 mutations across the 33 TCGA cancer types (Fig. 3B). In addition, we compared the mutation patterns in known cancer genes between matched primary and relapsed tumors and found that some somatic alterations in the primary tumors became lost in the relapsed tumors, which may be due to that clones bearing the alterations were sensitive to chemotherapy and killed; while relapsed tumors acquired many more mutations (Supplementary Fig. 3A), collectively increasing the complexity of treating relapsed tumors based on the molecular profiles of primary tumors.

To identify potential SCNA drivers, we first employed CNVkit[23] to perform SCNA segmentation based on the 24 primary tumor and normal sample pairs and then identified significant SCNA peaks using GISTIC2[24]. We detected 18 amplification peaks and 15 deletion peaks (FDR = $10^{-3}$, Fig. 3C). Focusing on a set of well-established SCNA driver genes[25], we found that ARID1A and PARK2 were the most frequently deleted genes, whereas ZNF217, CDKN1B, and KRAS were amongst the most frequent amplifications (Fig. 3D). Comparing the matched primary and relapsed tumors, we found frequent gain or loss of SCNA driver genes (Supplementary Fig. 3B). In addition, we performed an analysis on the loss of heterozygosity (LOH) and found frequent reciprocal LOH events, which is similar to the observation in TGCTs[11]. For example, chromosome 10 underwent chromosome-wide copy-number neutral LOH in 4 out of the 24 primary YSTs. These results provide a global view of the SCNA landscape in YSTs.

**Mutational patterns in YST patients with gonadal dysgenesis.** We identified two YST patients (i.e., P02 and P03) who were diagnosed with 46 XY pure gonadal dysgenesis. These patients had mal-developed gonads and malignant YST at the time of diagnosis (age at diagnosis: P02, 33; P03, 18). Both patients had

female external genitalia and primordial uterus. Based on WES data, we indeed detected a significant amount of reads mapped to chromosome Y-linked male-specific sex-determining gene, SRY (Fig. 4A, D), confirming their male-like chromosomal structures. Therefore, we focused on the tumor samples from these two patients for detailed investigations. Strikingly, the tumors of both patients harbored double mutations in KRAS and TP53 (P02, TP53 p.C176Y and p.G262D, KRAS p.P34L, Fig. 4B; P03, TP53 p. I225T and KRAS p.Q61R, Fig. 4E). Notably, KRAS p.P34L has been shown to be GAP-resistant and locked in the active state in a way similar to oncogenic KRAS p.G12V[26]; and the mutation at KRAS codon 61 can impair PASGAP-mediated GTP hydrolysis, leading to constitutive GTP binding and activation[27]. To specifically test the significance of TP53 and KRAS concurrent mutations, we performed an analysis using a pan-cancer cohort of 9125 TCGA patients and found only 2.1% (191 out of 9125) harbor mutations in both TP53 and KRAS simultaneously, suggesting that double mutated TP53 and KRAS genes tend to be enriched in the YST patients with gonadal dysgenesis (Fisher's exact test, $p = 5.3 \times 10^{-4}$). Furthermore, we noted that the P02 tumor harbored a highly deleterious frameshift mutation in the sex-determining gene[28], DMRT1 (Fig. 4B, Supplementary Fig. 4A–C), and the P03 tumor contained a point mutation in KCTD1 that is linked to Scalp-ear-nipple syndrome[29] in which patients often exhibit sex-related aberrations such as under-developed or absent nipples, and absent breast tissues (Fig. 4E, Supplementary Figure 4E–G). Compared to those in the primary tumors, the mutant allele frequencies of these genes were lower in relapsed tumors (paired test, Fig. 4B, E). These results provide molecular insights into cancers with abnormal development of the reproductive system and suggest that the coordination of classic cancer drivers and aberrations in sex-specific genes may play an important role in the tumor evolution of this special patient group (Fig. 4C, F and Supplementary Fig. 4D, H).

**Evolution of one YST with multiple relapsed samples.** To explore the evolutionary trajectory of metastatic progression in YST, we studied the clonal evolution from primary to relapsed tumors (Supplementary Fig. 5). In particular, we focused on one patient, P07, who had one primary (P07-P) and three relapsed tumor samples (P07-R1, P07-R2, and P07-R3). In this patient, a metastatic site in the transverse colon (P07-R1) was detected nine months after the first surgery, and seven months later, two other metastatic sites were found: one in the spleen (P07-R2), and one in the pelvic cavity (P07-R3) (Fig. 5A). An analysis of the mutational overlap showed that only two mutations (one was missense and the other was synonymous) were shared across the four tumor samples (Fig. 5B). Two independent phylogenetic analyses (Fig. 5C and Supplementary Fig. 6) consistently suggested that the late metastatic samples (P07-R2 and P07-R3)

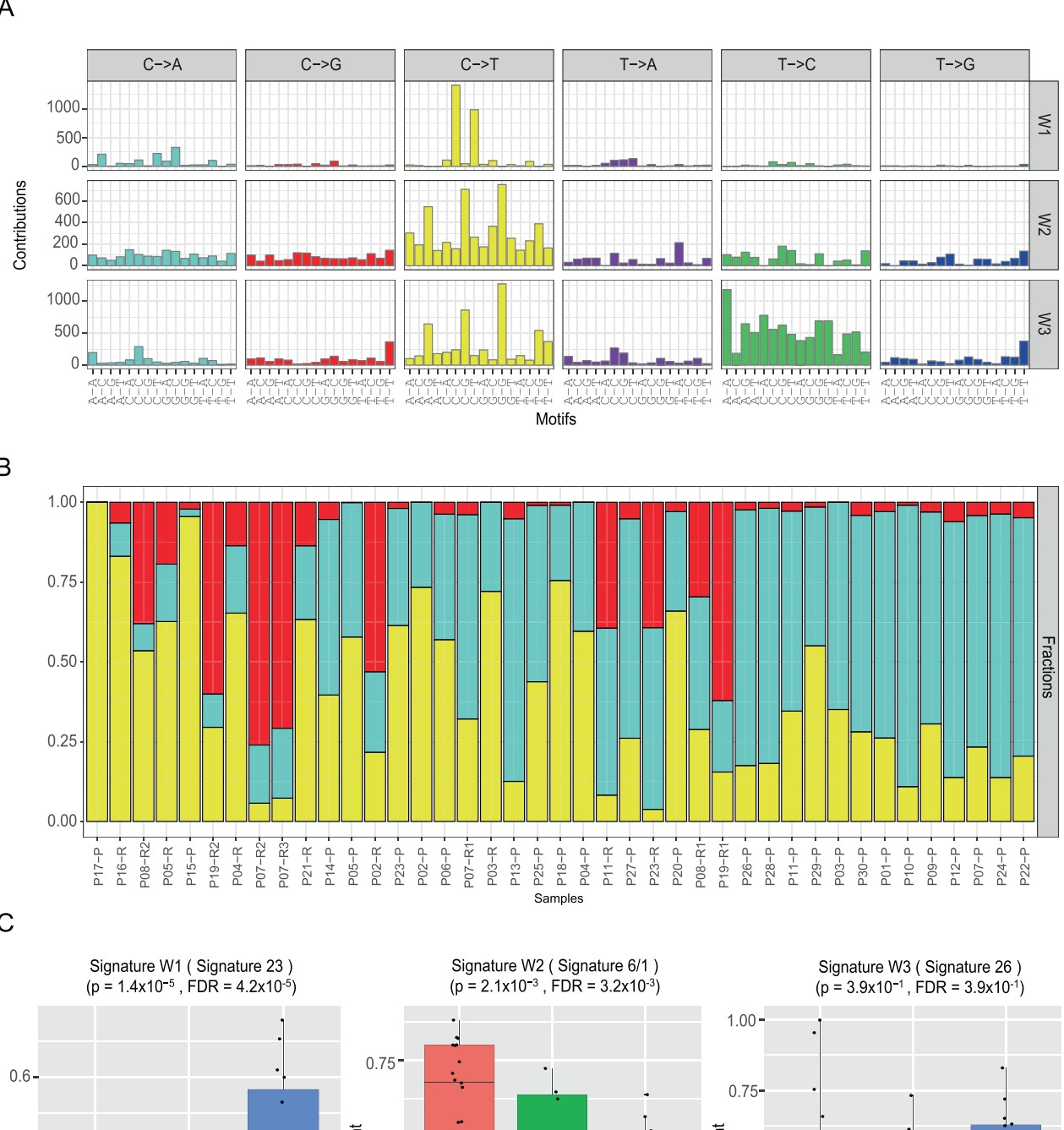

**Fig. 2 Mutational signatures of YST. A** Mutational signatures W1-W3. A Bayesian NMF approach was used to decompose the mutational spectrum across the 41 YST tumor samples into distinct mutational signatures. Each color refers to one of the six base substitutions, each of which is further stratified by the adjacent 5′ and 3′ flanking nucleotides. **B** The relative contributions of mutational signatures in each tumor sample. **C** The comparison of each signature among chemo-sensitive primary, chemo-resistant primary, and relapsed samples. *P* values were computed based on Kruskal–Wallis tests (primary_sensitive: $n = 19$ biologically independent samples; primary_resistant: $n = 7$; relapse: $n = 15$), and FDR were used for multiple testing corrections. The middle line in the box is the median, the bottom and top of the box are the first and third quartiles, and the whiskers extend to 1.5× interquartile range of the lower and the upper quartiles, respectively.

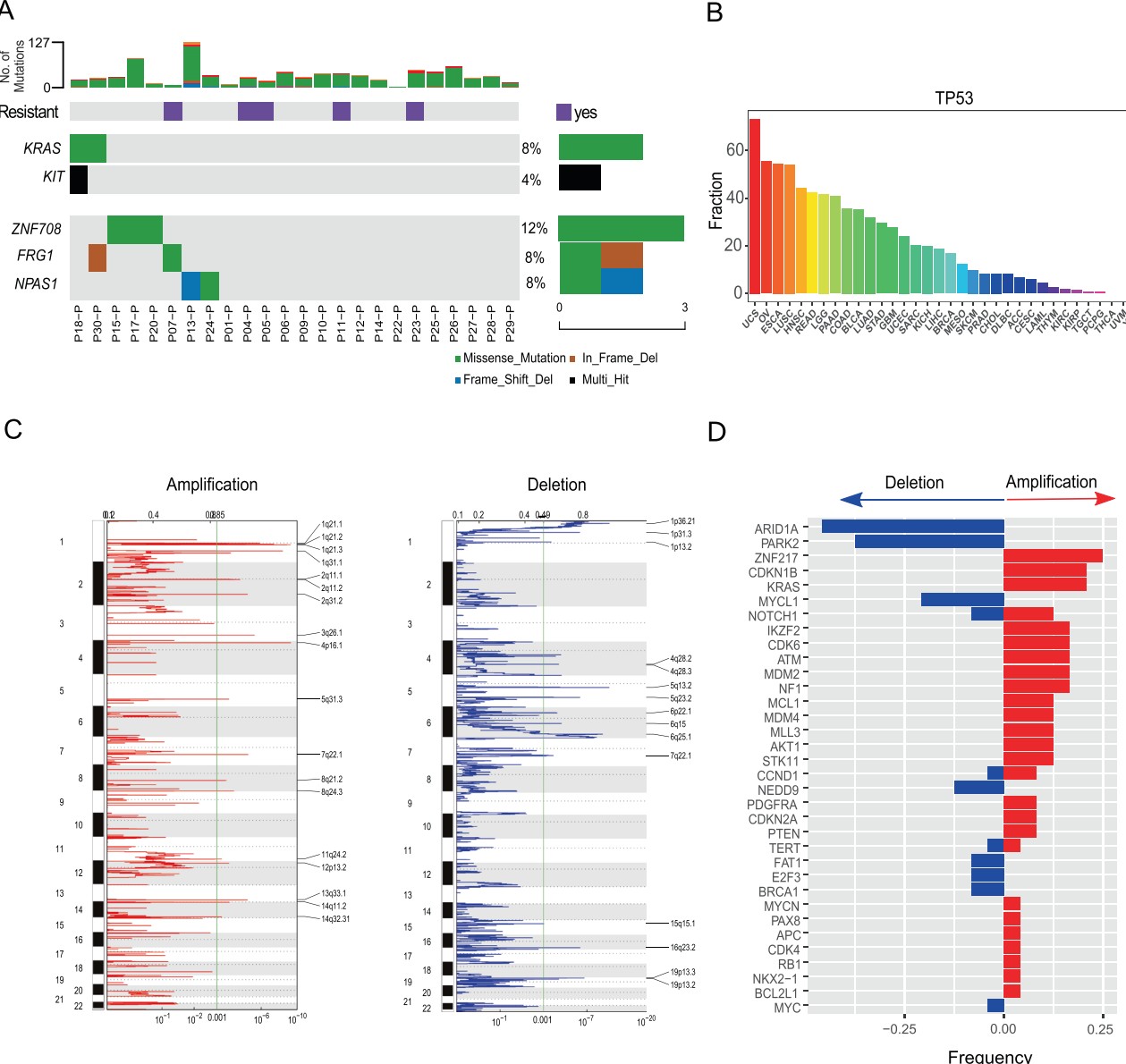

**Fig. 3 The somatic alteration patterns of YST primary tumors. A** Significantly mutated genes (SMGs) in all YST primary tumors ($n = 24$) except for those from patients P02 and P03, with gonadal dysgenesis. The bars on the top and the right show the mutational rate observed for each tumor and the composition of mutations in selected genes, respectively. Two known cancer driver genes are in bold, and three driver candidates were identified by both MuSiC2 and MutSigCV. **B** The comparison of *TP53* missense mutation frequencies in YST and 33 TCGA cancer types. **C** Amplification and deletion signals identified in YST. The segmentation data from the 24 YST primary samples were pooled for GISTIC2 analysis. Significantly amplified (left panel) or deleted (right panel) peaks were identified at FDR = $10^{-3}$. The top and bottom numerical values refer to G-scores and q values, respectively. **D** The copy-number alteration frequencies of cancer drivers across primary YST samples.

were derived from the primary tumor (P07-P), while the early metastatic sample (P07-R1) showed an earlier split from the common ancestor. We further confirmed the inferred evolutionary relationship by a LOH analysis showing that all the tumors except for P07-R1 shared the genome-wide LOH pattern (Fig. 5D). In addition to this case, tumors from several other patients shared no obvious driver events (e.g., P04 and P23, Supplementary Fig. 3A and B), implying a branched evolutionary model in YST. These results suggest that the YST evolution pattern could be more complicated than the conventional, linear progression view of primary → early metastasis → late metastasis.

**A role of *OVOL2* overexpression in chemoresistance**. To characterize the molecular mechanisms related to the chemoresistance of YSTs more comprehensively, we generated RNA-seq data for 12 YST samples (three sensitive primary tumors and nine relapsed tumors) (Supplementary Data 1). Through a comparison of gene expression profiles between sensitive and relapsed samples, we identified 153 significant differentially expressed genes (fold-change >2 and FDR <0.1; upregulated: 136; downregulated: 17) (Fig. 6A). These differential genes were not only enriched in KRAS signaling, but also pancreas beta cells, and bile acid metabolism (Fig. 6B). To further pinpoint potential contributors to the chemo-resistance, we correlated the expression levels of

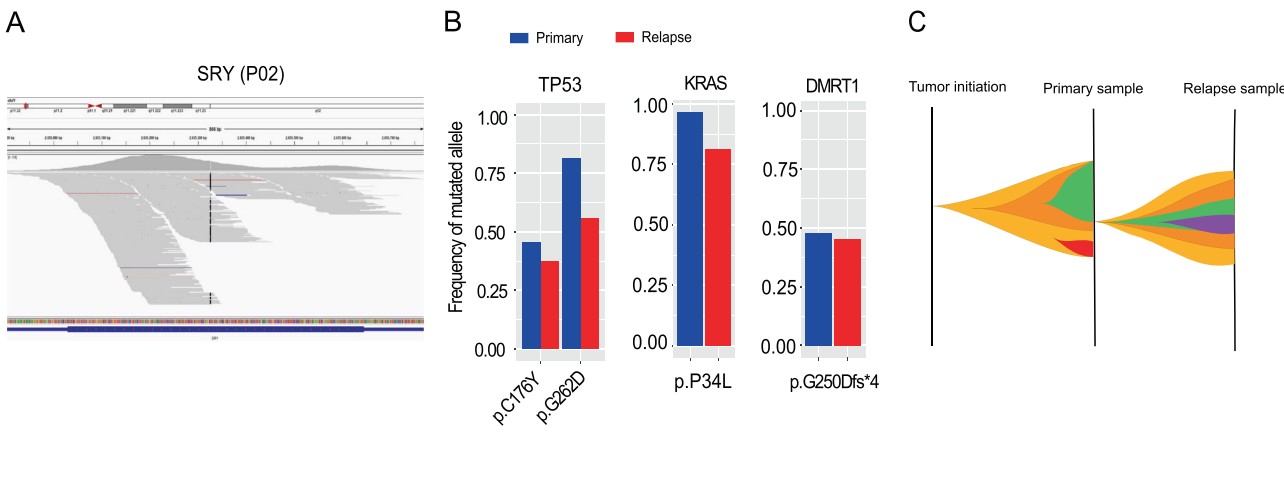

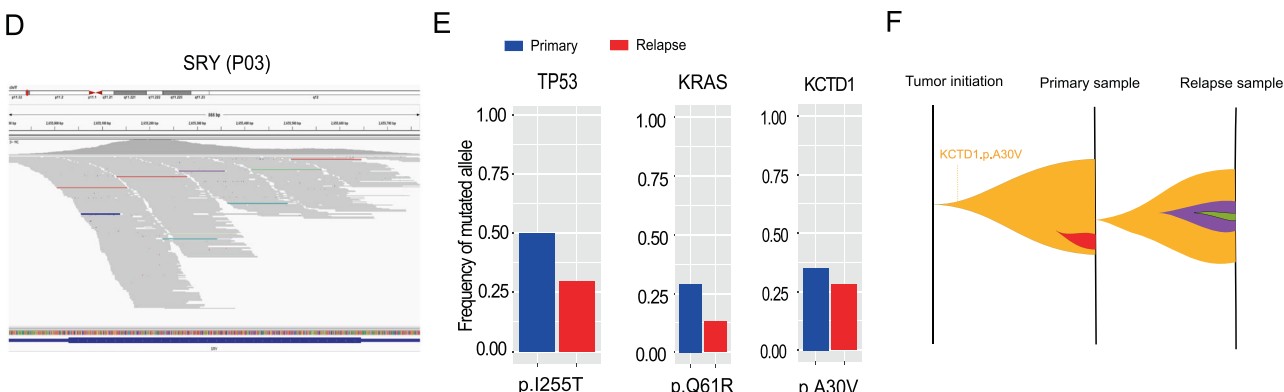

**Fig. 4 Key drivers and clonal evolution of YST patients with abnormal gonad development.** The whole-exome sequencing reads mapped to sex-determining region Y protein, SRY, from Y chromosome in the primary tumor sample of patients P02 (**A**), and P03 (**D**). The allele frequencies of *TP53, KRAS, DMRT1*, and *KCTD1* in primary and relapse samples from P02 (**B**), and P03 (**E**). The raw allele frequency was calculated based on the proportion of the reads containing the altered allele among all the reads mapped to that position. A schematic representation of the putative clonal evolution of YSTs in the patients, P02 (**C**), and P03 (**F**). Subclones are shown in different colors.

these differential genes with resistance to cisplatin (the chemotherapy standard for YST patients) using drug sensitivity data of cancer cell lines[30] (Fig. 6C). We found that *OVOL2* (ovo like zinc finger two) was a top candidate; this gene showed a significantly higher expression level in relapsed tumors than in sensitive tumors (Fig. 6C), and a high expression of *OVOL2* was associated with cisplatin resistance in cancer cell lines of different lineages (Wilcoxon rank-sum test, $p = 4.8 \times 10^{-9}$, Fig. 6D), and specifically, in ovarian cancer cell lines (Wilcoxon rank-sum test, $p = 6.7 \times 10^{-3}$, Fig. 6E). These results suggest that the over-expression of *OVOL2* may be related to the mechanism of chemoresistance in YSTs.

We next performed in vitro experiments to assess whether *OVOL2* expression plays a causal role in affecting sensitivity to cisplatin. We chose an ovarian YST cell line (NOY1), which showed higher *OVOL2* expression than the normal human ovarian surface epithelial cell line (HOSEpiC) (Fig. 7A, Supplementary Fig. 7). We silenced the *OVOL2* expression using two independent small interfering RNAs (siRNAs) (Fig. 7B) and found that, in NOY1, the knockdown of *OVOL2* dose- and time-dependently decreased cell viability in the presence of cisplatin (Fig. 7C, D). Furthermore, the silencing of *OVOL2* sensitized NOY1 to apoptosis (measured as Annexin V and propidium iodide double-stained cells) after cisplatin treatment (Fig. 7E, F, Supplementary Fig. 8). Consistent with these results, we found more caspase-3 cleavage in the *OVOL2*-knockdown samples (Fig. 7G, H, Supplementary Fig. 7). These data suggest that

*OVOL2* knockdown indeed sensitized NOY1 cells to cisplatin by enhancing apoptosis.

## Discussion

Through whole–exome sequencing of a well-annotated clinical tumor sample set, we characterized the mutational landscape of YST, a major subtype of ovarian germ cell tumors. Compared with other cancer types, YSTs appeared to have infrequent *TP53* mutations. Assessing clinical response, we showed that MSI were potentially informative in predicting resistance to chemotherapy. Mayer et al. first reported a positive correlation between MSI and chemotherapy resistance in TGCTs. They found that 45% patients with tumors in the refractory group showed MSI, while the rate was only 6% in the chemo-sensitive group[31]. Other studies consequently reported genetic instability presented as MSI might contribute to treatment refractory by chemotherapy resistance[32,33]. Interestingly, COSMIC signature 6, a signature due to defective DNA repair and often found in microsatellite unstable tumors did not show such a pattern; signature 1, which correlates with the age of diagnosis, showed a positive correlation with chemo-sensitivity. These results lay a foundation for developing predictive biomarkers in this disease, and further efforts should be made to validate these findings using independent patient cohorts.

We systematically identified mutated driver gene candidates, including known drivers *KRAS* and *KIT*, and candidates such as

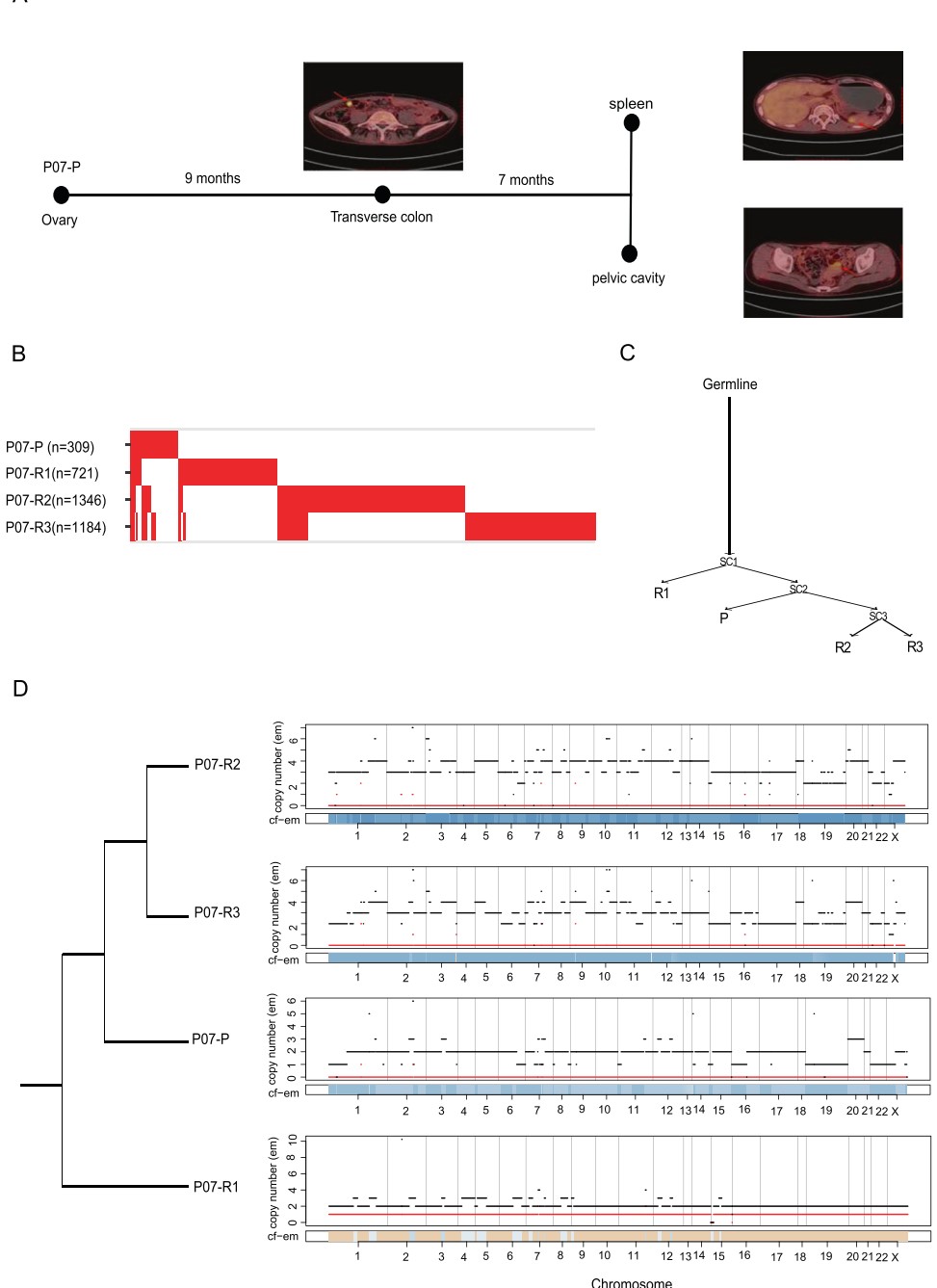

**Fig. 5 Evolution of YST progression from a single patient. A** The PET-CT tomography of patient P07 at different diagnostic time points. **B** The alignment of all mutations in the primary and three relapse tumor samples from P07. The mutation numbers included in the analysis are shown in parentheses. **C** A phylogenetic tree showing evolutionary relationships between the tumor samples. Selected mutations are shown along the lineages. **D** A phylogenetic tree among the four tumor samples from P07 based on their LOH patterns.

*FRG1* and *ZNF708*. *FRG1* (Facioscapulohumeral Muscular Dystrophy [FSHD] Region Gene 1) is an evolutionarily conserved and protein-coding gene. The function of *FRG1* includes regulating muscle development, angiogenesis[34,35]. Many studies have demonstrated its involvement in tumorigenesis and tumor progression: *FRG1* mutations occur in patients with thyroid cancer and calcifying fibrous tumor of pleura[36,37]. Functionally, *FRG1* may act as tumor suppress gene since its expression is reduced in various types of tumor, such as prostate, breast, lung, gastric, etc[38,39]. Further, *FRG1* knockdown can lead to significantly enhanced invasion and migration abilities of tumor cells[38].

*ZNF708* (Zinc Finger Protein 708) is a protein-coding gene and possibly involved in transcriptional regulation[40]. However, its role in tumor development has not been reported. Given the intact *TP53* gene in YSTs, one intriguing question is the somatic drivers underlying recurrent MSI. However, our analysis did not confidently identify candidates, and further efforts are warranted in this regard.

In parallel to whole-exome sequencing, we employed RNA-sequencing to characterize the transcriptomes of some primary and relapsed YST samples and identified a potential role of *OVOL2* in cisplatin resistance through influencing tumor

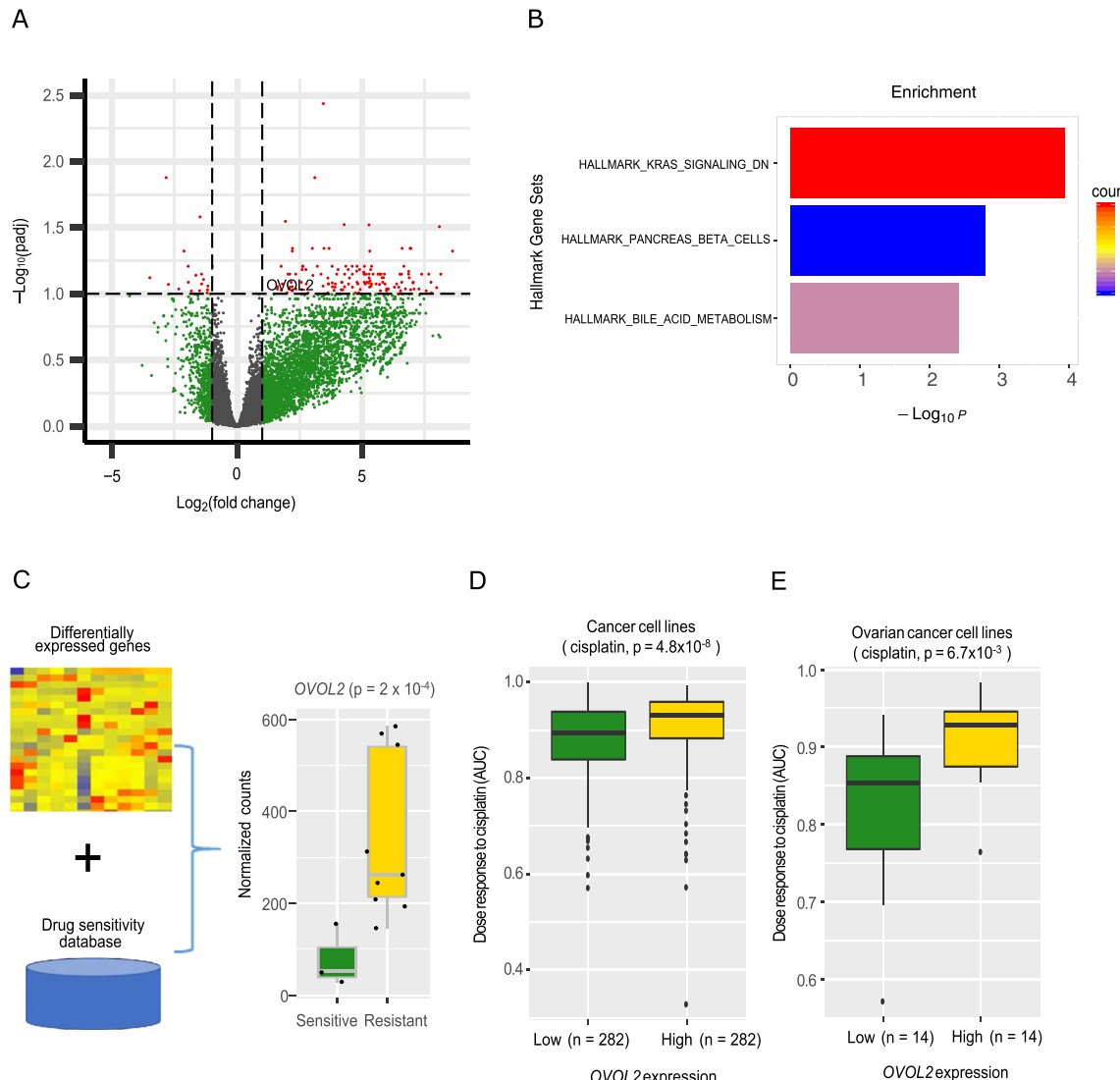

**Fig. 6 Differentially expressed genes between chemo-sensitive primary and relapsed YST samples. A** A volcano plot showing genes differentially expressed in primary and relapsed YST samples (fold-change >2, FDR <0.1). **B** Enrichment of differential genes in cancer hallmark pathways. **C** Correlation analysis of drug sensitivity and expression level identifies *OVOL2* as a top candidate. The left panel show normalized count numbers from sensitive ($n = 3$ biologically independent samples) and resistant ($n = 9$) groups. The $p$ value was computed by DEseq2. **D** The expression levels of *OVOL2* in cancer cell lines with different responses to cisplatin (two-sided Wilcoxon rank sum test, $p = 4.8 \times 10^{-8}$; *OVOL2* expression low group: $n = 282$ biologically independent samples; *OVOL2* expression high group: $n = 282$). **E** The expression levels of *OVOL2* in ovarian cancer cell lines with different responses to cisplatin (two-sided Wilcoxon rank sum test, $p = 6.7 \times 10^{-3}$; *OVOL2* expression low group: $n = 14$; *OVOL2* expression high group: $n = 14$). **C–E** The middle line in the box is the median, the bottom and top of the box are the first and third quartiles, and the whiskers extend to 1.5× interquartile range of the lower and the upper quartiles, respectively.

cellapoptosis. As an evolutionarily conserved transcription factor, *OVOL2* regulates angiogenesis, cranial neural tube and heart formation[41,42]. It also play a critical role in tumor progression through epithelial–mesenchymal transition (EMT)[43]. OVOL2 regulates EMT program and safeguards the epithelial cell type[44]. In epithelial tumors, such as colorectal cancer, breast cancer and lung adenocarcinoma, several studies have demonstrated that *OVOL2* suppress invasion capability by inhibiting EMT and OVOL2 protein expression level is related with patient's prognosis[45–47]. Further investigations can focus on several aspects: (i) through profiling additional primary resistant YSTs to figure out whether its overexpression is gained in response to treatment or inherent in resistant tumors; (ii) further validate its functional roles using additional cell lines and animal models; and (iii) elucidate its detailed mechanisms in the context of EMT.

Finally, our study investigated two patients with abnormal gonadal development, 46 XY karyotype. In contrast to the other YST patients, their tumors were characterized by double mutated *KRAS* and *TP53*, a rarely observed pattern. Due to the very small size ($n = 2$) included in the analysis, profiling additional such samples are necessary to confirm this pattern. People with sexual development disorders are at an increased risk of developing germ cell tumors[48], but such cancer patients have been understudied in the field. Given their distinct molecular aberrations, our results highlight a need to develop alternative treatment strategies for this special group of cancer patients.

## Methods
**Patient selection and sample cohort**. The 30 YST patients in this study were recruited at Peking Union Medical College Hospital, China. The study was conducted in accordance with the declaration of Helsinki and was approved by the

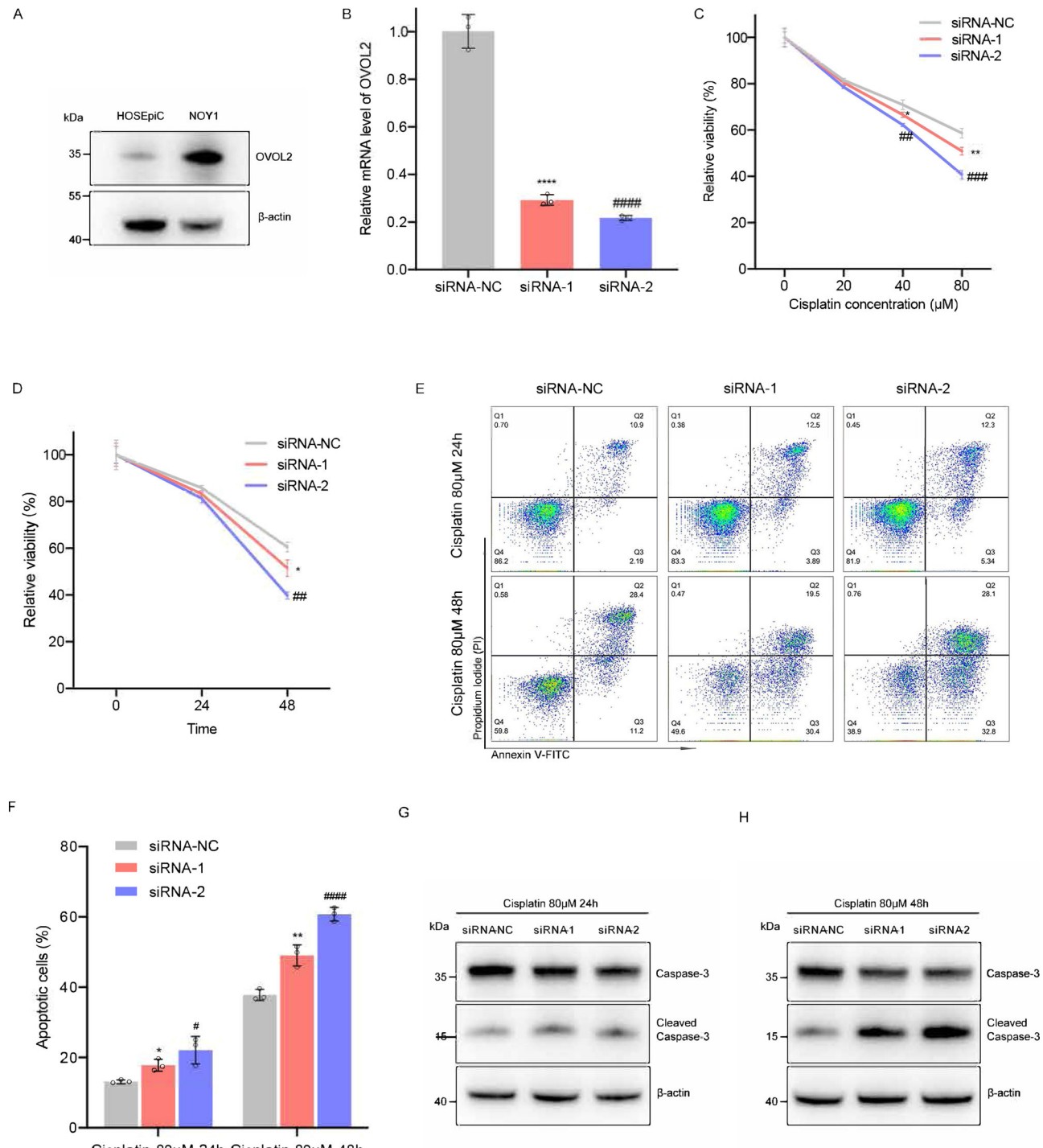

Ethics Committee of Peking Union Medical College Hospital, Beijing (project ID: JS–1747). All patients provided written informed consent before treatment, sample collection, and analysis. The pathological features of the 30 patients were strictly evaluated based on the standard criterion by multiple independent senior pathologists.

All the YST patients except for P14 and P22 received surgery, followed by standard platinum-based chemotherapy. Regimens for the primary treatment included bleomycin, etoposide, cisplatin, or bleomycin, vincristine, cisplatin; and repeated every 21 days. The patients received additional two consolidated cycles after AFP declining to normal ranges for initial treatment. The drug dosage was calculated according to the patient's body surface area. The recurrent treatment included salvage surgery and chemotherapy, and the regimens were chosen according to the National Comprehensive Cancer Network guideline. Since the tumor burdens for P14 and P22 were very high, chemotherapy was performed first before initiating standard treatment, as described above. A complete response to the initial treatment was defined as satisfactory surgical resection with negative tumor marker and an absence of radiographic evidence for tumor masses. Patients who received the initial treatment and achieved a complete response without any evidence of relapse for at least six months were classified into the chemo-sensitive group. Otherwise, the patients were classified into the chemo-resistant group. Only three patients (P21, P24, and P25) from the sensitive group relapsed more than six months after the standard treatment and received further cytoreductive surgery followed by salvage chemotherapy. All three patients lived without evidence of disease for at least two years. The fasting serum AFP level was measured by an electro-chemiluminescence assay (Roche Diagnostics, Shanghai, China). In the chemo-resistant group, two patients, P02 (age at diagnosis: 33 years old) and P03 (age at diagnosis: 18 years old), had never menstruated and were diagnosed with 46 XY pure gonadal dysgenesis. Based on the whole–exome sequencing data, we examined the read coverage of the sex-determining gene, *SRY*, which is located on the Y chromosome.

**Fig. 7 The effect of OVOL2 expression on the sensitivity of NOY1 cell lines to cisplatin. A** The protein level of OVOL2 in HOSEpiC and NOY1 cells by western blot. **B** The relative mRNA expression levels of *OVOL2* in control and siRNA-knockdown cells. $N = 3$ independent experiments. Two-tailed Student's *t*-test, siRNA-NC vs. siRNA-1: $p < 10^{-4}$, siRNA-NC vs. siRNA-2: $p < 10^{-4}$. **C** Drug response across different drug concentrations. NOY1 cells were transfected with siRNAs for 48 h and further treated with gradient concentration of cisplatin (0, 20, 40, 80 μM) for additional 48 h. Cell viability was detected by CellTiter-Glo assay and the relative viability was normalized to the respective noncisplatin-treated conditions. $N = 3$ independent experiments. Two-tailed Student's *t*-test, at 40 μM cisplatin treatment condition, siRNA-NC vs. siRNA-1: $p = 3.5 \times 10^{-2}$, siRNA-NC vs. siRNA-2: $p = 2.3 \times 10^{-3}$; at 80 μM cisplatin condition, siRNA-NC vs. siRNA-1: $p = 7.2 \times 10^{-3}$, siRNA-NC vs. siRNA-2: $p = 4.0 \times 10^{-4}$. **D** Drug response across different treatment time points. NOY1 cells were transfected with siRNAs for 48 h and further treated with 80 μM cisplatin. Cell viability was detected by CellTiter-Glo assay at 0, 24 and 48 h, and the relative viability was normalized to the respective 0 h conditions. $N = 3$ independent experiments. Two-tailed Student's *t*-test, siRNA-NC vs. siRNA-1; $p = 1.9 \times 10^{-2}$, siRNA-NC vs. siRNA-2: $p = 2.0 \times 10^{-4}$. **E–H** NOY1 cells were transfected with siRNAs for 48 h and further treated with 80 μM cisplatin for additional 24–48 h. Cell apoptosis and expression levels of marker proteins in apoptosis (Caspase-3 and cleaved Caspase-3) were determined by flow cytometry (E, F) and western blot (G, H). $N = 3$ independent experiments. Two-tailed Student's *t*-test, at 24 h, siRNA-NC vs. siRNA-1: $p = 1.0 \times 10^{-4}$, siRNA-NC vs. siRNA-2: $p < 10^{-4}$; at 48 h, siRNA-NC vs. siRNA-1: $p < 10^{-4}$, siRNA-NC vs. siRNA-2: $p < 10^{-4}$. For all in vitro experiments in this study, each one was repeated three times. (A, G and H), β-actin was used as internal control. Uncropped blots of figures are shown in Supplementary Fig. 7. (B, C, D and F), Error bars are presented as means ± s.d. *P* values were analyzed by two-tailed Student's *t*-test. */#$p < 0.05$, **/##$p < 0.01$, ***/###$p < 10^{-3}$, and ****/####$p < 10^{-4}$.

**Whole-exome and RNA-seq data generation**. Genomic DNA from FFPE tumor samples was extracted using GeneRead DNA FFPE kit (QIAGEN) or Maxwell® FFPE Plus DNA Kit (Promega); genomic DNA from fresh tumor samples was extracted using QIAamp DNA Mini Kit (QIAGEN); and blood samples of YST patients was extracted using TGuide Blood Genomic DNA Kit (TIANGEN). Then, 0.1–1 μg DNA was sheared into 200–300 bp fragments using a Covaris kit (Covaris, MA, USA). The resulting DNA fragments were repaired and 3′ A-tailed. Adapters were ligated to both ends of the fragments, followed by size selection. Size-selected fragments were amplified via polymerase chain reaction (PCR). Exome capture was performed using SureSelect Human All Exon V6 (Agilent) according to the manufacturer's protocol, followed by PCR amplification. Libraries were prepared using the KAPA Library Quantification Kit (Kapa Biosystems, MA, USA). Validated DNA libraries were sequenced on Illumina XTEN or NovaSeq 6000. For RNA sequencing, total RNA from 12 freshly frozen samples of 10 YST patients was extracted using the Trizol method. For each sample, 1 μg total RNA was used to generate the libraries using the NEBNext Ultra II RNA library Prep Kit (Illumina, USA). The libraries were sequenced on Illumina NovaSeq 6000, generating on average, 28 million 2 × 150 bp paired-end reads per sample.

**Somatic mutation rate and mutation signature analysis**. Whole-exome sequencing read pairs were trimmed and only read pairs with ≤15 N bases, and >50% high-quality bases were kept for subsequent analyses. The resulting high-quality reads were aligned to the human reference genome (Homo_sapiens_assembly19) using Burrows–Wheeler Aligner (0.7.17)[49]. To improve the alignment accuracy, we used Genome Analysis Toolkit (version 3.8.1)[50] to process BAM files, including marking duplicates, and local realignment around high-confidence insertion and deletions. Based on ~7000 high-frequency SNP sites, we confirmed that the matched primary tumor, relapsed tumor, and normal samples were indeed from the same patient using BAM-matcher[51] (identical genotype call ≥80%). To call somatic mutations accurately, we used the variant calling pipeline developed by TCGA MC3 project[16]. Briefly, this pipeline employs five callers to call substitution mutations, and three callers to identify small indels, with detailed annotation. We only kept substitution mutations and indels supported by at least two callers for further analyses. All the mutations were kept for subsequent analyses if the position was ≥10× in both normal and tumor samples. To fairly compare the mutation rate of YST with other cancer types, we employed a mutation dataset of 9125 samples from 33 cancer types that have been analyzed using the same TCGA MC3 pipeline[52]. To calculate the TMB values, we only used missense or ORF shift mutations in the overlapped targeted regions of SureSelect Human All Exon V6 used in this study and those defined in the TCGA MC3 project. For multiple tumor samples from the sample patient, we reconciled their mutation calls to increase the detecting sensitivity.

The mutational signature discovery was performed by SignatureAnalyzer based on the Bayesian variant of NMF[53]. Using mutation data across all 41 samples, we repeated the analysis 100 times and found that in >60% of the analysis, the mutation spectrum was decomposed into three mutational signatures, W1–W3. We also performed the analysis based on the well-defined mutation signatures by Catalogue Of Somatic Mutations In Cancer (COSMIC, https://cancer.sanger.ac.uk/cosmic/signatures_v2) via the tool, deconstructSigs[54]. To compare the mutational signatures, we classified all the YST samples into three groups: primary chemo-sensitive, primary chemo-resistant, and relapse. The Kruskal–Wallis test was used to test the contribution difference of the mutational signatures among the three groups, and FDR was used to correct for multiple testing effects. We used MANTIS[17] to call the MSI status for each tumor based on 2539 loci from the mSINGS package[55].

**Identification of significantly mutated genes**. We used MuSiC2[21] to identify significantly mutated genes (FDR < 0.2) using the primary samples (except samples

from the two patients with gonadal dysgenesis, P02, and P03). Among the 20 SMGs, two cancer genes were identified in a previous study[21]. To identify mutation drivers specific to YST, we further confirmed the SMG significance of the remaining 16 candidates using MutSigCV[22]. Maftools[56] was used to generate waterfall plots of the selected genes. The mutational frequency comparison of *TP53* was based on TCGA data using the same MC3 pipeline, and only mutations (SNVs and indels) resulting in amino acid changes or frameshifts were kept for this analysis.

**Somatic copy-number alteration and LOH analysis**. We performed SCNA analysis, in parallel to the mutation analysis, based on pooled segmentation data from YST tumor samples (samples from gonadal dysgenesis patients were excluded). Given paired tumor-normal whole-exome sequencing data, we first determined SCNAs using CNVkit (v0.9.3)[23] with default parameters. Then, based on the corresponding segmentation values, we used GISTIC2[24] to identify regions with a statistically significant frequency of copy-number alterations. To identify the potential SCNA drivers, we used the file, all_thresholded.by_genes.txt, generated by GISTIC2, which applied both low-level (cut-off, +/−1) and high-level (cut-off, +/−2) thresholds to define the gene-level SCNAs. We focused on a known set of genes comprised of frequently amplified oncogenes or deleted tumor suppressor genes, as previously defined[25]. and only high-level (+/−2) SCNAs were considered as amplification/deletion. To compare the pattern between primary and relapsed samples, we performed a similar analysis using all tumor samples except for the four samples from the two patients with gonadal dysgenesis. To perform LOH analysis, we first called germline variants through GATK[57]. The germline variants in the analysis met the three criterions: (i) covered by WES probes; (ii) depth ≥10; and (iii) supported reads ≥3. We employed snp-pileup to generate files in correct format to feed the LOH caller, FACETS[10]. We defined the chromosome-wide LOH if 90% of a specific chromosome underwent LOH.

**Clonal structure analysis**. To explore the clonal evolution of tumors from the two patients with gonadal dysgenesis, we inferred the dynamic changes of clonal structures using sciClone[58]. Based on segmentation output from the CNVkit v0.9.3 as described above, the high-confidence SCNAs were defined as those with segmentation values (log₂) >0.5 or <−0.5 and excluded from the clonal analysis. Variants located in LOH regions were also excluded. For P02, there were 1516 SNV mutations in at least one sample, and 1196 (79%) were kept for the clonal structural analysis after filtering. For P03, P04, P05, P11, and P23, the numbers were 58% (578/968), 8% (137/1784), 4% (99/2335), 46% (403/884), and 47% (613/1291).

**Phylogenetic tree construction**. To construct the evolutionary relationship between multiple tumor samples from the same patient, we employed Treeomics[59] to build the phylogenetic tree based on the nucleotides from all the sites that were identified as mutations in at least one tumor sample. We further confirmed, the result by a maximum parsimony analysis using MEGAX[60]. We performed a bootstrap analysis (1000 times) to estimate the statistical confidence for internal nodes.

**RNA-seq data analysis**. We used TopHat2[61] to align RNA-seq reads to the reference and quantified gene expression level through cufflinks (v2.2.1)[62]. To identify differentially expressed genes between primary (sensitive) and relapse (resistant) tumor samples, we used DESeq2[63] with the following criteria: absolute fold-change >2 and FDR <0.1 (raw *p*-value < $9 \times 10^{-4}$). A volcano plot was generated by the R package, EnhancedVolcano. To identify the genes that were likely to contribute to chemoresistance, we obtained cisplatin sensitivity and gene expression data of cancer cell lines from DepMap (https://depmap.org). For each of the 153 differentially expressed genes identified above, we assessed the correlation of its expression level and cisplatin sensitivity using Spearman rank correlation. We used clusterProfiler[64] to perform the gene set enrichment analysis with cancer

hallmark pathways downloaded from MSigDB (https://www.gsea-msigdb.org/gsea/msigdb/index.jsp) based on differentially expressed genes.

**Cell culture and treatment**. The human YST cell line (NOY1) was purchased from a laboratory at Nagoya University (Nagoya, Japan). NOY1 cells were cultured in Roswell Park Memorial Institute (RPMI) 1640 medium (Corning, USA) containing 10% fetal bovine serum (Gibco, USA). The human ovarian surface epithelial cells (HOSEpiC) were a gift from Dr. Keng Shen (Peking Union Medical College Hospital, Beijing, China) and were grown in Dulbecco's Modified Eagle Medium (Corning, USA), supplemented with 10% fetal bovine serum(Gibco, USA). All cells were kept in a humidified atmosphere containing 5% $CO_2$ at 37 °C.

**Transient transfection of siRNA**. NOY1 cells at 50–60% confluence were transfected with two different siRNAs targeting *OVOL2* mRNA and one nontargeting siRNA by Lipodectamin RNAiMAX reagent (Invitrogen, Cat. 13778150) according to the manufacturer's instructions. Briefly, NOY1 cells were grown in six-well plates and transfected with siRNA for 12 h and medium was replaced. After 48 h of transfection, real-time PCR was used to detect the knockdown efficiency.

**Cell viability assay**. The number of viable cells in culture was measured using the CellTiter-Glo One Solution Assay (Promega, USA) according to the manufacturer's protocol. Briefly, 48 h after siRNA transfection, $5 \times 10^3$ NOY1 cells were seeded into 96 wells plates and allowed to grow overnight. After 12 h of seeding, cells were treated with cisplatin at gradient concentrations for 24–48 h. At the end of the indicated treatment period, 100 μL CellTiter-Glo reagent was added and mixed with cultured cells by shaking 2 min, and then the plate was allowed to incubate at room temperature for 10 min. Luminescence data was recorded on a multi-plate reader (Envision). Relative viability was normalized to the respective cisplatin-treated condition at 0 h or noncisplatin-treated condition. Each sample condition contained three biological replicates, and the experiment was performed at least three technical replicates.

**Apoptosis assay**. The apoptosis of NOY1 cells was detected using Annexin-V, FITC Apoptosis Detection Kit (Dojindo, Japan). NOY1 cells were transfected with the siRNAs for 48 h and further treated with 80 μM cisplatin. At 24 h, 48 h after cisplatin treatment, NOY1 cells were harvested using trypsin digestion solution without EDTA (Solarbio, China) and washed with PBS twice. And then the cells were resuspended in binding buffer and incubated in the dark with FITC Annexin-V and propidium iodide (PI) for 15 min at room temperature. Stained cells were analyzed on an Attune$^{TM}$ NxT flow cytometer (Thermo Fisher Scientific, USA). Data analysis was performed using the FlowJo 10 software.

**Total RNA isolation and real-time PCR**. After the siRNA transfection, total RNA from NOY1 cells was extracted using Trizol reagent (Invitrogen, USA). Then, 1 μg RNA was reverse-transcribed to cDNA using a reverse transcription kit (TaKaRa, Japan). The cDNA was amplified on a CFX96 Real-Time PCR Detection System (Bio-Rad, USA) using SYBR Green PCR Master Mix (TaKaRa, Japan). The expression of β-actin was used to normalize the expression level of *OVOL2*. Relative mRNA expression was calculated using the $2^{-\Delta\Delta CT}$ method. Each sample condition contained three biological replicates, and data was presented as mean ± standard deviation (S.D.). The experiment was performed at three technical replicates. The sequences of primers used in this study were listed as follows: *OVOL2*, forward, 5′-ACAGGCATTCGTCCCTACAAA-3′, reverse, 5′-CGCTGCTTATAGGCATACTGC-3′; and β-actin, forward, 5′-GAGAAAAT CTGGCACCACACC-3′, reverse, 5′-GGATAGCACAGCCTGGATAGCAA-3′.

**Western blot analysis**. Cells were lysed in RIPA buffer (Beyotime, China) with freshly added 0.01% protease inhibitors (Roche) and then centrifuged at $15,000 \times g$ at 4 °C for 15 min. The supernatants were harvested and quantified using the BCA protein assay kit (Beyotime, China) according to the manufacturer's instructions. Then 20 μg total proteins were applied and resolved on 4–12% SDS-PAGE gradient gels and transferred to 0.22 μm PVDF membranes. Membranes were blocked with 5% nonfat milk and then incubated with the following primary antibodies: caspase-3 (Abcam, ab3235, 1:1000), Cleaved Caspase-3 (Abcam, ab32042, 1:500), OVOL2 (Abcam, ab1-69469, 1:500), and β-actin (Abcam, ab8226, 1:500). The membranes were incubated at 4 °C overnight and then washed by TBST solution for three times. After that, they were applied with appropriate secondary antibodies. Signals were detected using electrochemiluminescence methodology. All experiments were repeated at least three times.

**Statistical analysis**. For in vitro experiments, all values were presented as mean ± S.D. Statistical significance was determined using two-tailed Student's *t*-test, and differences between means with $P < 0.05$ were considered significant. Statistical analysis and graph-plotting were performed with Prism 5.0 software (GraphPad Software, San Diego, CA).

**Reporting summary**. Further information on research design is available in the Nature Research Reporting Summary linked to this article.

## Data availability

The sequencing data reported in this study have been deposited and are publicly accessible through the following repositories: the Genome Sequence Archive of the Beijing Institute of Genomics Data Center (http://bigd.big.ac.cn/gsa, accession number HRA000131); NCBI Gene Expression Omnibus [https://www.ncbi.nlm.nih.gov/geo, accession number GSE169733]; and the European Variation Archive ([https://www.ebi.ac.uk/ena/browser/view/PRJEB44264]). The remaining data are available in the Article file, Supplementary Information or available from the authors upon request. https://www.ebi.ac.uk/ena/browser/view/PRJEB44264.

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

## Acknowledgements

This study is supported by Chinese Academy of Medical Sciences Initiative for Innovative Medicine (CAMS-I2M_1_002). We gratefully acknowledge the support from the Suzhou New District, Jiangsu Province, China. We thank Kamalika Mojumdar for editorial assistance.

## Author contributions

H. Liang and J.Y. conceived of and designed the research. X.Z. and Y.Z. contributed to the collection of specimens and discussion of clinical significance; X.P. led the data analysis; H. Li and X.G. contributed to the data analysis; D.C., M.Y., and J.W. contributed to the collection of specimens and performed the experiments. X.P., X.Z., H. Liang, and J.Y. wrote the manuscript with input from all authors. H. Liang supervised the whole project.

## Competing interests

X.P., H. Li, and X.G. are full-time employees of Precision Scientific; and H. Liang is an advisor and shareholder of Precision Scientific. All other authors declare no competing interest.
