## [Peer Review File · Nature Communications]

Reviewers' comments:

Reviewer #1 (Remarks to the Author): Expertise in germ cell tumours and chemosensitivity

In this manuscript, Zong et al. describe the first genomic analysis of yolk sac tumors (YSTs). Clinical tumor samples with distinct responses to cisplatin-based chemotherapy were analyzed by whole-exome and RNA sequencing, which revealed somatic driver candidates and copy-number alteration drivers. Differential expression analysis of sensitive versus relapsed tumors and in vitro knock-down experiments suggest that *OVOL2* overexpression may be related to chemoresistance in YSTs. This study is exciting, as it is the first insight into the genomic landscape of YSTs, which are a rare but understudied tumor type affecting young adults. However, there is an overall lack of discussion of results and their significance as well as some experimental design limitations and lack of detail. Specific issues that should be addressed are as follows:

The authors state in several places (including the abstract) that 'YSTs appear to have the lowest TP53 mutation frequency across all tumors'. TP53 mutations are extremely rare in other types of GCTs, and because the authors have sequenced only 24 tumors, it is not clear that the TP53 mutation frequency is actually different from the low levels documented in other GCTs. It is appropriate for the authors to conclude that TP53 mutations are very infrequent in YSTs but they should be more cautious about saying that the frequency is the lowest among all cancer types.

TGCTs have been found by others to show reciprocal LOH, with gains of one parental allele and simultaneous loss of the other (ref 11 cited by the authors). Does this also occur in YSTs?

The authors observe *OVOL2* overexpression in relapsed vs. sensitive tumors. Does the *OVOL2* locus undergo mutation or copy number alterations in the tumors that show elevated expression? The functional validation of *OVOL2* would have been strengthened by assessment in additional cell lines.

The discussion is quite limited and underdeveloped. For instance, the authors report differences in frequency of specific base alterations in YSTs; what is the significance of this observation and what does it tell us about mechanism? Likewise, the authors report that relapsed tumors have much higher TMB. Is this related to chemotherapy exposure and is that reflected in the mutational signature? There is no discussion of the novel driver candidates *ZNF708* and *FRG1*—what is their known function? How could they be driving tumorigenesis? Also there is no description of what is known about *OVOL2* function or how *OVOL2* overexpression might be contributing to chemoresistance.

Many of the figure panels are not explained clearly in the legends. For example, in Fig 1D, what is the blue line and what is the shaded area? In Fig 1E are the dotted line connecting matched samples? In Fig 3C what are the numerical values at the top and bottom of the plots? Fig 4 includes complex representations of data that receive only a cursory description in the figure legend.

The availability of the primary data (eg RNA-seq data sets) is unclear.

Minor points:

Patient treatment was referred to as “standard cisplatin-based chemotherapy”, but should be defined further. Were the patients only receiving cisplatin, or in combination with other drugs or adjuvants? Were they all on the same dose and treatment schedule?

The authors report two different Kras mutations in YST patients with gonadal dysgenesis. One is in the frequently mutated codon 61. The other is a P34L change—has that also been implicated in malignant transformation in other studies?

In the section on mutational signatures (p 5), the sentence about cosine similarity analysis lists signature 6 twice, with different (undefined) numerical values associated with each.

Reviewer #2 (Remarks to the Author): expertise in germ cell tumours genomics and transcriptomics

Zong and colleagues explores the genomic changes of yolk sac tumors and determine if DNA or RNA features can identify features causing sensitivity to carboplatin. They performed WGS on 41 samples from 30 patients and RNA sequencing on 12 patients with known sensitivity to platinum.

In the comparison to TGCT, did you use all TGCT or did you compare to just the yolk sac subset. It might be more appropriate to subset out the TGCT to just the yolk sac cohort as well for a more appropriate comparison. Genes that were listed in figure 3, KIT and KRAS, were not actually observed mutated in the yolk sac tumors profiled there, only in the seminomas. But KRAS copy number was high in the yolk sac tumors.

The primary-resistant samples had a higher MSI score, but were lower for the contribution of Signature W2 and E3 which were attributed to Cosmic Signature 6 which is associated with DNA repair and MSI tumors. It seems like this should be commented on in the discussion.

In figures S3A and S3B, it showed that a lot of the events in the primary were not visibly in the recurrences. So I was curious about the clonal heterogeneity of these samples. Yet, in the section on evolution, you only highlight one. I think it would be important to at least mention that the others had a similar trend and to comment more on this in the discussion.

For Figures 4B and E, did you adjust the mutation allele frequency for the tumor purity? The clonality data in figures 5C and F could also use additional detail. For P03, are you sure that the KRAS and KCTD1 mutations are in the same clone as the TP53 mutation. The lower MAF suggests it is more a subclone.

Two of the patients with KRAS mutations are not in hotspot regions. Do they still show activated KRAS signaling?

The gene pathway enrichment gene sets seemed odd to find pancreas and bile acid sets. Was this

evaluated more? Was this because the recurrences were in non-ovary sites and included several colon and intestine?

Do you have RNA available to go back to the other YST tumors and run rtPCR to see if the OVOL2 expression is gained in response to treatment or is inherent and expressed in the primary tumors of patients that were resistant?

AFP was found to be different at 6 months point treatment for patients that were sensitive to treatment versus those that were resistant. Did you have baseline levels of AFP to determine if there was a difference prior to treatment? Or is this more of a marker to establish response at the 6 month period used to call samples sensitive or resistant?

The discussion is really limited and could benefit from additional information expanding on some of the observations that were identified in the results.

For the clonal structural analysis, how many regions were thrown out because of the SCNAs? Were the majority for the SNVs able to be used for the SciClone analysis?

The paper is currently missing the accession number for the raw sequencing data.

For Figure 1E, the TMB for the relapse samples were averaged. Was the TMB fairly consistent across the relapses when more than 1 was profiled? Considering you think the evolution process is not linear, it could suggest a big difference in span of TMB.

For figure 5 B, it would help to have some context of number of mutations, boxes or a designated n per sample. Maybe also comment on the mutations that were shared across all tumors? Also what role did copy number play in the clonal heterogeneity since Figure S3B also showed large differences between the primaries and the recurrences.

I didn't see any data showing the differences between sensitive and recurrent samples for the SCNA data, but it was mentioned in the methods.

It would be beneficial to add what the current known function of OVOL2 and why it is thought to be involved with resistance to cisplatin.

Reviewer #3 (Remarks to the Author): expertise in phylogenetics

Zong et al. perform a genomic study of yolk sac tumors, a relatively rare but understudied form of cancer. There is a reasonable argument for the importance of a study like this in potentially improving treatment of an underserved patient population. The fact that the genomics of these tumors are unusual in some ways also makes them of interest for basic cancer research. For example, it is interesting that TP53 mutation is rare in these tumors and that makes them worth studying in itself, although it is not obvious we get too much insight here into why these tumors are

unusual in that regard. It is similarly interesting that they frequently show microsatellite instability (MSI) and that this might be a biomarker for resistance, although again there is no obvious answer emerging here why these tumors in particular often exhibit MSI. The authors have managed to recruit a good-sized cohort for a study of a relatively rare cancer like this, sufficiently powered for at least some useful analysis. The two-person cohort of patients with abnormal gonad development is not large enough for any real statistical analysis but this is understandable given their rarity and I think the paper is enriched by the analysis of these cases even if it is just anecdotal. The methodology appears sound, with a few possible exceptions noted below. In particular, the sequencing and data analysis pipelines appear appropriate and well designed, again with a few possible exceptions noted below.

My concerns and criticisms are as follows:

1. The mutational signature analysis is a nice addition and provides some mechanistic insight into the mutational landscape of these tumors, but raised a few questions about methodology. I found it curious that two of the inferred signatures are mapped to the same Sanger signature. Are there notable differences between them? It may be preferable to attempt a signature analysis fitting to the known COSMIC signature catalog without de novo signature discovery, to remove potential for ambiguity in deconvolution into signatures that comes when one infers novel signatures.
2. In regards to signature analysis, I am also curious why the text commented on interpretation of signatures 6 and 23, but not on 26, which also is implicated with DNA mismatch repair defects.
3. Related to the preceding points, I would question the assertion that W1 is significantly higher in sensitive tumors since the p-value (0.047) is at best weakly significant and not even that if one controls for multiple hypothesis testing, e.g., that each signature would be tested for relationship to sensitivity.
4. I also question the conclusion that double mutation of TP53 and KRAS is significant in the two patients with abnormal gonad development. I believe that test would also need to be controlled for multiple hypotheses. Any pair of genes might by chance have come up as mutated in both, and so the test should control for the numbers of pairs of genes that might have been considered, which I believe would render it non-significant even if we assume the authors only looked at pairs of known high-frequency cancer drivers.
5. The phylogenetic analysis is a valuable addition for the paper, but I do have concerns about the methodology. The provided examples of mutant allele frequencies seem to suggest that there are differences in purity or subclonal heterogeneity between the primary site and recurrence, which is not so surprising. If I understand correctly, though, there is no control for this in the evolutionary tree building, which is just maximum parsimony based on presence or absence of missense mutations. Clonal heterogeneity can confuse this kind of tree building, so I would suggest inferring clonal structure before tree-building or using a tool specific for tumor phylogenetics that integrates clonal deconvolution with tree-building, e.g., PhyloSub, CITUP, or AncesTree.
6. It is unfortunate that there is only one patient for which the data is amenable to phylogenetic

analysis. Tumor phylogenies tend to be quite variable from patient to patient and without looking at multiple tumors one cannot easily appreciate what is a robust feature of the progression process vs. what is just a chance effect of a single tumor. It might be worth considering a comparable analysis for at least the tumors with two samples (primary plus recurrence or two recurrence) even if just to test whether the observation of branched rather than linear trajectory is consistent across tumors.

7. As noted above, it is interesting that these tumors exhibit some unusual genetic features (lack of TP53 loss and frequently MSI). I therefore found it disappointing that no insight into the mechanisms behind these features seemed to emerge from the genomic data. Am I correct that none of the significantly mutated or copy number altered genes was suggestive of a possible driver of recurrent MSI in these tumors? And did anything emerge as an obvious alternative to TP53 as a source of frequent SCNA in these tumors? If there were relevant results there I missed, I think that would be worth at least mentioning in the discussion.

I also noted a couple of minor errors:

- In the Abstract, “copy number alternation” should be “copy number alteration”
- On p. 3, “reticular-microcystic pattern, and is” should be “reticular-microcystic pattern, and this is”?

Point-by-point Response

Reviewer #1

In this manuscript, Zong et al. describe the first genomic analysis of yolk sac tumors (YSTs). Clinical tumor samples with distinct responses to cisplatin-based chemotherapy were analyzed by whole-exome and RNA sequencing, which revealed somatic driver candidates and copy-number alteration drivers. Differential expression analysis of sensitive versus relapsed tumors and in vitro knock-down experiments suggest that *OVOL2* overexpression may be related to chemoresistance in YSTs. This study is exciting, as it is the first insight into the genomic landscape of YSTs, which are a rare but understudied tumor type affecting young adults.

However, there is an overall lack of discussion of results and their significance as well as some experimental design limitations and lack of detail.

Response: Thank you for this comment. We have substantially extended the discussion about our results, their significance and limitations (Page 9-11).

The authors state in several places (including the abstract) that ‘YSTs appear to have the lowest TP53 mutation frequency across all tumors’. TP53 mutations are extremely rare in other types of GCTs, and because the authors have sequenced only 24 tumors, it is not clear that the TP53 mutation frequency is actually different from the low levels documented in other GCTs. It is appropriate for the authors to conclude that TP53 mutations are very infrequent in YSTs but they should be more cautious about saying that the frequency is the lowest among all cancer types.

Response: We agree. We have changed the related text to “YSTs appear to have infrequent *TP53* mutations” (Page 2 and Page 9).

TGCTs have been found by others to show reciprocal LOH, with gains of one parental allele and simultaneous loss of the other (ref 11 cited by the authors). Does this also occur in YSTs?

Response: Per suggestion, we have performed a LOH analysis based on our WES data. We did observe frequent reciprocal LOH events, and for example, chr10 shows chromosomal-wide copy-number neutral LOH in 4 out of the 24 primary YSTs. We have included these results on Page 7.

The authors observe *OVOL2* overexpression in relapsed vs. sensitive tumors. Does the *OVOL2* locus undergo mutation or copy number alterations in the tumors that show elevated expression?

Response: We have examined the 12 tumor samples included in our expression analysis, and found no protein-changing mutations detected in *OVOL2*. We also checked the copy-number data of these samples and found that the median value for the resistant group was higher than that of sensitive group but not statistically significant (Wilcoxon rank sum test, $p = 0.3$).

The functional validation of OVOL2 would have been strengthened by assessment in additional cell lines.

Response: We agree that functional validation in additional cell lines would strength our results. Unfortunately, for YST, NOY1 is the only YST cell line we can find for this disease. We have added some texts to discuss this limitation (Page 10).

The discussion is quite limited and underdeveloped. For instance, the authors report differences in frequency of specific base alterations in YSTs; what is the significance of this observation and what does it tell us about mechanism?

Response: We have significantly extended our discussion on Page 9-10.

Likewise, the authors report that relapsed tumors have much higher TMB. Is this related to chemotherapy exposure and is that reflected in the mutational signature?

Response: This is a very good point. As for mutational signatures, the contributions of signature 4 (aetiology, exposure of tobacco mutagens) and signature 23 (aetiology, unknown) were much higher in relapsed tumors than primary tumors, and this pattern may be due to chemotherapy exposure. We have included these results in Figure S2E and Page 4, 6.

There is no discussion of the novel driver candidates ZNF708 and FRG1—what is their known function? How could they be driving tumorigenesis?

Response: Thank you for this comment. Currently, we don't know the exact mechanisms on how ZNF708 and FRG1 mutations contribute to tumorigenesis in this disease. But we have added some discussion about the biological roles of these two genes in Discussion (Page 10)

Also there is no description of what is known about OVOL2 function or how OVOL2 overexpression might be contributing to chemoresistance.

Response: We have added the following text related to function and the possible mechanism to chemoresistance on Page 10.

Many of the figure panels are not explained clearly in the legends. For example, in Fig 1D, what is the blue line and what is the shaded area? In Fig 1E are the dotted line connecting matched samples? In Fig 3C what are the numerical values at the top and bottom of the plots? Fig 4 includes complex representations of data that receive only a cursory description in the figure legend.

Response: We have added more details in figure legends in this revised version. Specially, the blue line and the shaded area represent the regression line and the related 95% confidence interval, respectively. In Fig. 1E, the dotted lines indicate the matched sample pairs. In Fig. 3C, the numerical values at the top and bottom of the plots refers to G-scores and q-values, respectively.

The availability of the primary data (eg RNA-seq data sets) is unclear.

Response: We have deposited all the WES and RNA-seq data into Genome Sequence Archive (GSA, <https://bigd.big.ac.cn/gsa/>) under the accession number: HRA000131. We have added the information on Page 18.

Minor points:

Patient treatment was referred to as “standard cisplatin-based chemotherapy”, but should be defined further. Were the patients only receiving cisplatin, or in combination with other drugs or adjuvants? Were they all on the same dose and treatment schedule?

Response: We have added more detailed description on Page 12 as followings: “Regimens for the primary treatment included bleomycin, etoposide, cisplatin, or bleomycin, vincristine, cisplatin; and repeated every 21 days. The patients received additional two consolidated cycles after AFP declining to normal ranges for initial treatment. The drug dosage was calculated according to the patient’s body surface area. The recurrent treatment included salvage surgery and chemotherapy, and the regimens were chosen according to NCCN guideline.”

The authors report two different Kras mutations in YST patients with gonadal dysgenesis. One is in the frequently mutated codon 61. The other is a P34L change—has that also been implicated in malignant transformation in other studies?

Response: Based on FATHMM prediction, both of P34L and Q61R in KRAS are pathogenic (scores: 0.99 and 0.98). There are 4 records for KRAS^{P34L} and 150 for KRAS^{Q61R} in COSMIC database (<https://cancer.sanger.ac.uk/cosmic>). We have added the related literature to the text (Page 7).

In the section on mutational signatures (p 5), the sentence about cosine similarity analysis lists signature 6 twice, with different (undefined) numerical values associated with each.

Response: We fixed the typo in the revised version.

Reviewer #2

Zong and colleagues explores the genomic changes of yolk sac tumors and determine if DNA or RNA features can identify features causing sensitivity to carboplatin. They performed WGS on 41 samples from 30 patients and RNA sequencing on 12 patients with known sensitivity to platinum.

In the comparison to TGCT, did you use all TGCT or did you compare to just the yolk sac subset. It might be more appropriate to subset out the TGCT to just the yolk sac cohort as well for a more appropriate comparison. Genes that were listed in figure 3, KIT and KRAS, were not actually observed mutated in the yolk sac tumors profiled there, only in the seminomas. But KRAS copy number was high in the yolk sac tumors.

Response: Thank you so much for the suggestion. In the original manuscript, we used all TGCT for the comparison. In this revised version, we further compared the TMB difference from TGCT-seminoma, TGCT-nonseminoma and YST. We found no significant difference between seminoma and nonseminoma subtypes of TGCT ($p = 0.72$); and TMB of both subsets are significantly lower than that of YST (TGCT_non-seminoma vs. YST: $p = 5.5 \times 10^{-10}$; TGCT_seminoma vs. YST: $p = 3.8 \times 10^{-10}$). We did observe 4 out of 24 primary YST tumors with a significant copy-number gain of KRAS. We have included these results in Page 4.

The primary-resistant samples had a higher MSI score, but were lower for the contribution of Signature W2 and E3 which were attributed to Cosmic Signature 6 which is associated with DNA repair and MSI tumors. It seems like this should be commented on in the discussion.

Response: Signature W2 is actually a mixture of signature 1 and signature 6. Per this suggestion, we did the parallel analysis using known signatures. The difference of signature 6 between sensitive and resistant primary tumors are not significant ($p = 0.07$). We have added some comments on this (Page 10).

In figures S3A and S3B, it showed that a lot of the events in the primary were not visibly in the recurrences. So I was curious about the clonal heterogeneity of these samples. Yet, in the section on evolution, you only highlight one. I think it would be important to at least mention that the others had a similar trend and to comment more on this in the discussion.

Response: In the evolutionary section, we have focused on one with multiple relapsed tumor samples. Per this suggestion, we have added more discussion (Page 6).

For Figures 4B and E, did you adjust the mutation allele frequency for the tumor purity?

Response: No, we did not. Per this suggestion, we have tried three different tools, including ABSOLUTE, FACETS and Control-FREEC, to estimate tumor purity. Unfortunately, different tools gave very different values. Considering the high uncertainty of tumor purity estimation based on WES data, we decided to keep the raw allele frequencies here. We have clarified this in the figure legend (Page 24).

The clonality data in figures 4C and F could also use additional detail. For P03, are you sure that the KRAS and KCTD1 mutations are in the same clone as the TP53 mutation. The lower MAF suggests it is more a subclone.

Response: Because the TP53 mutations were affected by copy number alteration, it was excluded from the clonal analysis. We previously marked it in that way as it is usually a founder mutation. Now we have removed the TP53 mutations from the figure (Figure 4).

Two of the patients with KRAS mutations are not in hotspot regions. Do they still show activated KRAS signaling?

Response: We identified two amino acid-changing mutations, KRAS^{P34L} and KRAS^{Q61R}. KRAS^{P34L} has been shown to be GAP-resistant and locked in the active state in a fashion similar to oncogenic KRAS^{G12V} (Gremer et al., Hum Mutat. 2011, 32(1):33-34). The substitution at codon 61 in KRAS can impair PASGAP-mediated GTP hydrolysis, leading to constitutive GTP binding and activation (Sheffels et al., Science Signaling, 2018, 11:eaar8371). So, it is very likely that both KRAS mutations result in activated KRAS signaling. We have cited these references in the text (Page 7).

The gene pathway enrichment gene sets seemed odd to find pancreas and bile acid sets. Was this evaluated more? Was this because the recurrences were in non-ovary sites and included several colon and intestine?

Response: This is a great explanation. Indeed, several recurrent samples were from colon; and one was from intestine. That is why there is some lineage-specific signature. We have added this information (Page 9).

Do you have RNA available to go back to the other YST tumors and run rtPCR to see if the OVOL2 expression is gained in response to treatment or is inherent and expressed in the primary tumors of patients that were resistant?

Response: Unfortunately, we do not have RNA available in additional YST tumors. We have discussed this point (Page 10-11).

AFP was found to be different at 6 months point treatment for patients that were sensitive to treatment versus those that were resistant. Did you have baseline levels of AFP to determine if

there was a difference prior to treatment? Or is this more of a marker to establish response at the 6 month period used to call samples sensitive or resistant?

Response: AFP is an established specific tumor marker for yolk sac tumors. The AFP level in patients with yolk sac tumor is highly elevated before surgery, and decreases into the normal range after treatment. During the follow-up period, AFP and ultrasound are usually used for monitoring tumor recurrence. In general, the elevated AFP level appears earlier than the recurrent tumor detected by imaging (ultrasound, CT or PET-CT). The chemo-resistance in our study was defined as tumor recurrence occurs within six months after primary treatment. So, the AFP level at the six-month point is significantly higher in the chemo-resistant group than that in the chemo-sensitive group. Whether the preoperative AFP level is an independent prognostic factor of patients with yolk sac tumor is still controversial, so there is no explicit cutoff value to classify patients into two groups before treatment.

The discussion is really limited and could benefit from additional information expanding on some of the observations that were identified in the results.

Response: We have substantially extended the Discussion part on Page 9-11.

For the clonal structural analysis, how many regions were thrown out because of the SCNAs? Were the majority for the SNVs able to be used for the SciClone analysis?

Response: For the clonal structural analysis, we excluded the variants affected by copy-number alteration or located in regions with loss of heterogeneity (LOH) in either samples of the same patients. We still obtained sufficient mutations for clonal analysis. In Figure 4, for P02, there were 1516 SNVs present in at least one sample and 1196 (79%) were kept for the clonal structural analysis after filtering; for P03, 58% of the mutations were kept (578/968).

The paper is currently missing the accession number for the raw sequencing data.

Response: We have deposited the raw sequencing data to Genome Sequence Archive (GSA: <https://bigd.big.ac.cn/gsa/>) with the accession number HRA000131. We updated the information on Page 18.

For Figure 1E, the TMB for the relapse samples were averaged. Was the TMB fairly consistent across the relapses when more than 1 was profiled?

Response: For Figure 1E, we compared the TMB difference between primary and relapse tumor samples. There were six patients with both primary and relapsed samples. Only one patient, P07, had multiple relapsed samples. The TMB value for P07's three relapsed samples is 0.38, 5.16 and 4.00 (mean: 3.18; standard deviation: 2.49). But because only data point was affected by this issue, the general pattern we observed remains robust.

Considering you think the evolution process is not linear, it could suggest a big difference in span of TMB. For figure 5 B, it would help to have some context of number of mutations, boxes or a designated n per sample. Maybe also comment on the mutations that were shared across all tumors?

Response: The figure below shows the numbers of missense mutations and total SNVs used for the phylogenetic tree. There are only one missense mutation (CFAP54^{V2358A}) and one synonymous mutation (SERPINI2^{S323S}) shared by all of the four tumor samples from the patient, P07. We have added the mutation numbers to Figure 5 and added the comment to that section (Page 8).

Also what role did copy number play in the clonal heterogeneity since Figure S3B also showed large differences between the primaries and the recurrences. I didn't see any data showing the differences between sensitive and recurrent samples for the SCNA data, but it was mentioned in the methods.

Response: Figure S3B aims to show SCNAs for some well-known copy-number driver genes. Due to the small sample size and the limitation of inferring copy-number alterations from WES, there could be quite a bit of SCNA uncertainty so we decided not to go further.

It would be beneficial to add what the current known function of OVOL2 and why it is thought to be involved with resistance to cisplatin.

Response: We have added some discussion about the role and potential mechanism of OVOL2 (Page 10-11).

Reviewer #3 (Remarks to the Author): expertise in phylogenetics

Zong et al. perform a genomic study of yolk sac tumors, a relatively rare but understudied form of cancer. There is a reasonable argument for the importance of a study like this in potentially improving treatment of an underserved patient population. The fact that the genomics of these tumors are unusual in some ways also makes them of interest for basic cancer research. For example, it is interesting that TP53 mutation is rare in these tumors and that makes them worth studying in itself, although it is not obvious we get too much insight here into why these tumors are unusual in that regard. It is similarly interesting that they frequently show microsatellite instability (MSI) and that this might be a biomarker for resistance, although again there is no obvious answer emerging here why these tumors in particular often exhibit MSI. The authors have managed to recruit a good-sized cohort for a study of a relatively rare cancer like this, sufficiently powered for at least some useful analysis. The two-person cohort of patients with abnormal gonad development is not large enough for any real statistical analysis but this is understandable given their rarity and I think the paper is enriched by the analysis of these cases even if it is just anecdotal. The methodology appears sound, with a few possible exceptions noted below. In particular, the sequencing and data analysis pipelines appear appropriate and well designed, again with a few possible exceptions noted below.

My concerns and criticisms are as follows:

1. The mutational signature analysis is a nice addition and provides some mechanistic insight into the mutational landscape of these tumors, but raised a few questions about methodology. I found it curious that two of the inferred signatures are mapped to the same Sanger signature. Are there notable differences between them?

Response: Sorry. Signature appeared twice due to a typo in the previous version. To obtain more robust mutation signatures, we have extended our mutation set beyond the coding regions and updated the analysis. We identified three de novo signatures W1, W2, and W3, which corresponds to signature 23, signature 6/1 and signature 26, respectively. We have updated the results (Page 5) and Figure 2.

It may be preferable to attempt a signature analysis fitting to the known COSMIC signature catalog without de novo signature discovery, to remove potential for ambiguity in deconvolution into signatures that comes when one infers novel signatures.

Response: In this revised version, we performed the analysis as suggested using the tool, deconstructSigs (Rosenthal et al., Genome Biology, 2016, 17:16). We found the signature 1, 3, 4, 6, and 23 showed significant difference across the three tumor groups (FDR < 0.15). We have added the results (Page 5-5, Fig. S2E).

2. In regards to signature analysis, I am also curious why the text commented on interpretation of signatures 6 and 23, but not on 26, which also is implicated with DNA mismatch repair defects.

Response: We have added the interpretation of signature 26 on Page 5.

3. Related to the preceding points, I would question the assertion that W1 is significantly higher in sensitive tumors since the p-value (0.047) is at best weakly significant and not even that if one controls for multiple hypothesis testing, e.g., that each signature would be tested for relationship to sensitivity.

Response: In the revised version, based on the updated analysis, we have adjusted the p-values of mutation signatures for multiple testing (Figure 2C and Figure S2E). We also emphasized the importance of using independent cohorts to validate our findings in Discussion (Page 10).

4. I also question the conclusion that double mutation of TP53 and KRAS is significant in the two patients with abnormal gonad development. I believe that test would also need to be controlled for multiple hypotheses. Any pair of genes might by chance have come up as mutated in both, and so the test should control for the numbers of pairs of genes that might have been considered, which I believe would render it non-significant even if we assume the authors only looked at pairs of known high-frequency cancer drivers.

Response: Because of the very small size of tumors with abnormal gonad development, we did not even consider testing mutation combinations across different driver genes. Our hypothesis of TP53 and KRAS mutation combination came from the intriguing observation of these two samples. Therefore, only this specific hypothesis was tested. We have clarified this in the text (Page 7) and also emphasized the importance of using independent cohorts to validate our findings in Discussion (Page 11).

5. The phylogenetic analysis is a valuable addition for the paper, but I do have concerns about the methodology. The provided examples of mutant allele frequencies seem to suggest that there are differences in purity or subclonal heterogeneity between the primary site and recurrence, which is not so surprising. If I understand correctly, though, there is no control for this in the evolutionary tree building, which is just maximum parsimony based on presence or absence of missense mutations. Clonal heterogeneity can confuse this kind of tree building, so I would suggest inferring clonal structure before tree-building or using a tool specific for tumor phylogenetics that integrates clonal deconvolution with tree-building, e.g., PhyloSub, CITUP, or AncesTree.

Response: Thank you for the suggestion. We have updated the phylogenetic tree of the four samples from P07. We used two different methods to reconstruct the phylogenetic tree, including Maximum Parsimony from MEGAX (Kumar et al., Molecular Biology and Evolution, 2018, 35:1547-1549) and Treeomics (Reiter et al., Nature Communications 8, 2017, 14114), and obtained the same tree topology. To get more independent support for our inferred evolutionary relationship, we employed another tool, FACETS (Shen and Seshan et al., 2016, Nucleic Acids Res, 44(16):e131), to estimate the loss of heterogeneity (LOH) status of the four samples from the patient, P07. The tree based on the LOH information further confirms the phylogenetic topology. We have included the results in Figure 5 and Page 8.

6. It is unfortunate that there is only one patient for which the data is amenable to phylogenetic analysis. Tumor phylogenies tend to be quite variable from patient to patient and without looking at multiple tumors one cannot easily appreciate what is a robust feature of the progression process vs. what is just a chance effect of a single tumor. It might be worth considering a comparable analysis for at least the tumors with two samples (primary plus recurrence or two recurrence) even if just to test whether the observation of branched rather than linear trajectory is consistent across tumors.

Response: Thank you for your comment. In this study, except for P07 with one primary and three relapsed samples, we had only six patients with one primary and one relapsed samples. The primary and relapsed tumors from P02 and P03 shared TP53 and KRAS mutations, supporting the linear trajectory model, while P04 and P23 shared no obvious mutations, supporting the branched models. We have added the results on Page 8.

7. As noted above, it is interesting that these tumors exhibit some unusual genetic features (lack of TP53 loss and frequent MSI). I therefore found it disappointing that no insight into the mechanisms behind these features seemed to emerge from the genomic data.

Response: We have added some related discussion on Page 10.

Am I correct that none of the significantly mutated or copy number altered genes was suggestive of a possible driver of recurrent MSI in these tumors? And did anything emerge as an obvious alternative to TP53 as a source of frequent SCNA in these tumors? If there were relevant results there I missed, I think that would be worth at least mentioning in the discussion.

Response: Based on current results, we cannot confidently claim that any mutation or copy number alteration contribute to MSI in YST. We have added some related discussion on Page 10.

I also noted a couple of minor errors:

- In the Abstract, “copy number alternation” should be “copy number alteration”

Response: Fixed (Page 2).

- On p. 3, “reticular-microcystic pattern, and is” should be “reticular-microcystic pattern, and this is”?

Response: Fixed (Page 3).

REVIEWERS' COMMENTS:

Reviewer #1 (Remarks to the Author):

Overall, the manuscript has been significantly improved by the addition of new analyses, clearer explanations of the data shown, and additional discussion. However, as noted below there are still several instances where the interpretation of the results should be described more clearly.

The authors describe YST in two patients with 46,XY pure gonadal dysgenesis, and in both cases they observe both Kras and P53 mutations. In the discussion, the authors appropriately point out that the results come from only two patients and need to be confirmed in additional patients. However, the abstract suggests that in general YST from patients with abnormal gonadal development have Kras and p53 mutations. The wording in the abstract should be changed to something along the lines of 'Interestingly, YSTs appeared to have very infrequent TP53 mutations, whereas the tumors from two patients with abnormal gonadal development were characterized by both KRAS and TP53 mutations as well as abnormal sex-specific genes.'

The authors have added new data on reciprocal LOH in YST. When describing those findings, it would be appropriate to cite the study (Taylor-Weiner, et al., reference 11) that reported this type of alteration in TGCT.

On page 6, the authors compare mutations in matched primary and relapsed tumors and find that 'some somatic alterations in the primary tumors became lost in the relapsed tumors...' It is not clear to this reader how the authors interpret this. Are the mutations undergoing reversion? Is there heterogeneity in the primary tumor that is manifesting upon relapse? The text could be clearer on this point.

On page 9, the authors note that differentially expressed genes in sensitive vs. relapsed tumors included an enrichment of genes involved in pancreatic beta cells and bile acid metabolism. It is not clear what the authors mean when attributing this to 'some lineage-specific effects of relapsed tumors.'

On page 10, middle of page, the authors state about ZNF708 that "However, its role in tumor development have been reported" but probably mean to say that "However, its role in tumor development has NOT been reported"

Figures 1C and 3A lack y-axis labels.

Reviewer #2 (Remarks to the Author):

Thank you for addressing my comments. I have a few more based on the response.

For the analysis comparing TMB to TCGA, addressing histology was useful. But it is still unclear why you didn't directly compare to just the yolk sac tumors of TCGA for a direct comparison. TCGA's supplemental figure 2A showed highest TMB in yolk sac tumors (PMID: 29898407). The TCGA non-seminomas still contain a mix of embryonal, teratomas, yolk sac and mixed tumors.

For the KRAS mutations, was there also RNA expression data for those samples that you could use to determine if KRAS expression is upregulated as well.

For the GSEA bile acid and pancreatic sets. It might be better to word that not as lineage which suggests the cells originated from there from, but surrounding tissue from a metastatic site.

I think it would be important to add to the methods, the number of mutations retained after excluding variants in CNAs.

The updated Figure 5 phylogenetic tree for SNVs and LOH both put the first relapse as not a descendent of the primary tumor. The updated text does not fully vibe with the three trees in Figure 5 and S6.

Reviewer #3 (Remarks to the Author):

The authors have addressed all of my substantive criticisms of the prior manuscript and I find their responses satisfactory. As before, I find this a useful paper on an interesting and understudied cancer system, of interest to basic cancer research and of potential translational value. While the rarity of this particular cancer results in small sample sizes that limit the depth of the statistical analysis, what is presented is sound and the authors are able to reach some novel and intriguing conclusions from the data. I would therefore judge it to be a novel, scientifically significant, and technically sound paper.

My one remaining very minor point is that LOH is defined in a couple of places, on pages 7 and 8, as "loss of heterogeneity". I believe LOH is more commonly defined as "loss of heterozygosity", which I think is what the authors mean here.

Point-by Point Response

Reviewer #1:

Overall, the manuscript has been significantly improved by the addition of new analyses, clearer explanations of the data shown, and additional discussion. However, as noted below there are still several instances where the interpretation of the results should be described more clearly.

The authors describe YST in two patients with 46,XY pure gonadal dysgenesis, and in both cases they observe both Kras and P53 mutations. In the discussion, the authors appropriately point out that the results come from only two patients and need to be confirmed in additional patients. However, the abstract suggests that **in general YST from patients with abnormal gonadal development have Kras and p53 mutations**. The wording in the abstract should be changed to something along the lines of ‘ **Interestingly, YSTs appeared to have very infrequent TP53 mutations, whereas the tumors from two patients with abnormal gonadal development were characterized by both KRAS and TP53 mutations as well as abnormal sex-specific genes.**’

Response: Thank you for this comment.

The authors have added new data on reciprocal LOH in YST. When describing those findings, it would be appropriate to cite the study (Taylor-Weiner, et al., reference 11) that reported this type of alteration in TGCT.

Response: We have added this citation (P7).

On page 6, the authors compare mutations in matched primary and relapsed tumors and find that ‘some somatic alterations in the primary tumors became lost in the relapsed tumors...’ It is not clear to this reader how the authors interpret this. Are the mutations **undergoing reversion**? Is there **heterogeneity** in the primary tumor that is manifesting upon relapse? The text could be clearer on this point.

Response: The treatment killed the clones sensitive to chemotherapy, which was likely to be responsible for the observation that some somatic alterations in the primary tumors become lost in the relapsed tumors. We clarified this point on P6.

On page 9, the authors note that differentially expressed genes in sensitive vs. relapsed tumors included an enrichment of genes involved in pancreatic beta cells and bile acid metabolism. It is not clear what the authors mean when attributing this to ‘some lineage-specific effects of relapsed tumors.’

Response: We have removed this text (P9).

On page 10, middle of page, the authors state about ZNF708 that "However, its role in tumor development have been reported" but probably mean to say that "However, its role in tumor development has NOT been reported"

Response: We have corrected this bug (P10).

Figures 1C and 3A lack y-axis labels.

Response: We have added y-axis labels in Figure 1C and Figure 3A in the revised version.

Reviewer #2:

Thank you for addressing my comments. I have a few more based on the response.

For the analysis comparing TMB to TCGA, addressing histology was useful. But it is still unclear why you didn't directly compare to just the yolk sac tumors of TCGA for a direct comparison. TCGA's supplemental figure 2A showed highest TMB in yolk sac tumors (PMID: 29898407). The TCGA non-seminomas still contain a mix of embryonal, teratomas, yolk sac and mixed tumors.

Response: In this analysis, we used the clinical and mutation data from TCGA PanCan Atlas. Based the related clinical information, only those two subtypes ((seminoma and nonseminoma) are available for TGCT.

For the KRAS mutations, was there also RNA expression data for those samples that you could use determine if KRAS expression is upregulated as well.

Response: Thank you for the suggestion. Unfortunately, there is no RNA available to validate this.

For the GSEA bile acid and pancreatic sets. It might be better to word that not as lineage which suggests the cells originated from there from, but surrounding tissue from a metastatic site.

Response: We have removed the text (P9).

I think it would be important to add to the methods, the number of mutations retained after excluding variants in CNAs.

Response: We have added those numbers as suggested (P15).

The updated Figure 5 phylogenetic tree for SNVs and LOH both put the first relapse as not a descendent of the primary tumor. The updated text does not fully vibe with the three trees in Figure 5 and S6.

Response: We have revised the text to clarify the point (P8).

Reviewer #3

The authors have addressed all of my substantive criticisms of the prior manuscript and I find their responses satisfactory. As before, I find this a useful paper on an interesting and understudied cancer system, of interest to basic cancer research and of potential translational value. While the rarity of this particular cancer results in small sample sizes that limit the depth of the statistical analysis, what is presented is sound and the authors are able to reach some novel and intriguing conclusions from the data. I would therefore judge it to be a novel, scientifically significant, and technically sound paper.

My one remaining very minor point is that LOH is defined in a couple of places, on pages 7 and 8, as "loss of heterogeneity". I believe LOH is more commonly defined as "loss of heterozygosity", which I think is what the authors mean here.

Response: Thank you so much. We have changed "loss of heterogeneity" into "loss of heterozygosity".